
# FaIRv2.0.0: a generalised impulse-response model for climate uncertainty and future scenario exploration

Nicholas J. Leach[1], Stuart Jenkins[1], Zebedee Nicholls[2,3], Christopher J. Smith[4,5], John Lynch[1], Michelle Cain[1], Tristram Walsh[1], Bill Wu[1], Junichi Tsutsui[6], and Myles R. Allen[1,7]

[1]Department of Physics, Atmospheric, Oceanic, and Planetary Physics, University of Oxford, United Kingdom.
[2]Australian–German Climate and Energy College, University of Melbourne, Australia.
[3]School of Earth Sciences, University of Melbourne, Australia.
[4]School of Earth and Environment, University of Leeds, Leeds, UK.
[5]International Institute for Applied Systems Analysis, Laxenburg, Austria.
[6]Environmental Science Laboratory, Central Research Institute of Electric Power Industry, Abiko-shi, Japan.
[7]Environmental Change Institute, University of Oxford, Oxford, UK.

**Correspondence:** Nicholas J. Leach (nicholas.leach@stx.ox.ac.uk)

**Abstract.** Here we present an update to the FaIR model for use in probabilistic future climate and scenario exploration, integrated assessment, policy analysis and education. In this update we have focussed on identifying a minimum level of structural complexity in the model. The result is a set of six equations, five of which correspond to the standard Impulse Response model used for greenhouse gas (GHG) metric calculations in the IPCC's fifth assessment report, plus one additional physically-motivated additional equation to represent state-dependent feedbacks on the response timescales of each greenhouse gas cycle. This additional equation is necessary to reproduce non-linearities in the carbon cycle apparent in both Earth System Models and observations. These six equations are transparent and sufficiently simple that the model is able to be written in standard tabular data analysis packages, such as Excel; increasing the potential user base considerably. However, we demonstrate that the equations are flexible enough to be tuned to emulate the behaviour of several key processes within more complex models from CMIP6. The model is exceptionally quick to run, making it ideal for integrating large probabilistic ensembles. We apply a constraint based on the current estimates of the global warming trend to a one million member ensemble, using the constrained ensemble to make scenario dependent projections and infer ranges for properties of the climate system. Through these analyses, we reaffirm that simple climate models (unlike more complex models) are not themselves intrinsically biased "hot" or "cold": it is the choice of parameters and how those are selected that determines the model response, something that appears to have been misunderstood in the past. This updated FaIR model is able to reproduce the global climate system response to GHG and aerosol emissions with sufficient accuracy to be useful in a wide range of applications; and therefore could be used as a lowest common denominator model to provide consistency in different contexts. The fact that FaIR can be written down in just six equations greatly aids transparency in such contexts.





## 1 Introduction

Earth System Models (ESMs) are vital tools for providing insight into the drivers behind Earth's climate system, as well as projecting impacts of future emissions. Large scale multi-model studies, such as the Coupled Model Intercomparison Projects (Eyring et al., 2016; Taylor et al., 2012, CMIPs), have been used in many reports to produce projections of what the future climate may look like based on a range of different concentration scenarios, with associated emission scenarios and socio-
economic narratives quantified by Integrated Assessment Models (IAMs). In addition to simulating both the past and possible future climates, these CMIPs extensively use idealised experiments to try to determine some of the key properties of the climate system, such as the equilibrium climate sensitivity [ECS, Collins et al. (2013)], or the transient climate response to cumulative carbon emissions (Allen et al., 2009, TCRE).

While ESMs are integral to our current understanding of how the climate system responds to GHG and aerosol emissions, and provide the most comprehensive projections of what a future world might look like, they are so computationally expensive that only a limited set of experiments are able to be run during a CMIP. This constraint on the quantity of experiments necessitates the use of simpler models to provide probabilistic assessments and explore additional experiments and scenarios. These models, often referred to as simple climate models (SCMs), are typically designed to emulate the response of more
complex models. In general, they are able to simulate the globally averaged emission → concentration → radiative forcing → temperature response pathway, and can be tuned to emulate an individual ESM (or multi-model-mean). In general, SCMs are considerably less complex than ESMs: while ESMs are three dimensional, gridded, and explicitly represent dynamical and physical processes, therefore outputting many hundreds of variables, SCMs tend to be globally averaged (or cover large regions), and parameterise many processes, resulting in many fewer output variables. This reduction in complexity means that
SCMs are much quicker than ESMs in terms of runtime: most SCMs can run tens of thousands of years of simulation per minute on an "average" personal computer, whereas ESMs may take several hours to run a single year on hundreds of super-computer processors; and are much smaller in terms of the number of lines of code: SCMs tend to be on the order of thousands of lines, ESMs can be up to a million lines (Alexander and Easterbrook, 2015).

Several simple climate models are available, such as the two used in the Intergovernmental Panel on Climate Change (IPCC) Special Report on 1.5°C warming (IPCC, 2018, SR15): FaIR v1.3 (Smith et al., 2018) and MAGICC6 (Meinshausen et al., 2011a). However, while these models are "simple" in comparison to the ESMs they emulate, they are often still not so simple as to allow new users to gain enough familiarity with the underlying equations to understand their behaviour without significant effort. This learning curve reduces their uptake by the wider community, and has resulted in different research groups generally
using the single model that they are most familiar with (Nicholls et al., 2020) from the wide range of SCMs. In the past, this has led to a different simple models being used by different working groups in major reports, reducing the consistency of the overall work. We believe one key step towards a transparent and coherent process in IPCC Assessments would be the use of at least one common SCM as widely as possible throughout all working groups, allowing results to be directly comparable. Such





use would provide additional context alongside domain specific models. For this to be realised, an SCM that is both easy to
understand and adapt is required.

An important innovation of the IPCC 5th Assessment Report (Myhre et al., 2013) was the introduction of a fully transparent
set of equations (the AR5-IR model) for use in the calculation of GHG metrics. However, that model was not quite adequate
to reproduce the evolution of the integrated impulse response to emissions over time, due to the lack of non-linearity in the
carbon cycle. The Finite amplitude Impulse Response (FaIR) model v1.0 (Millar et al., 2017) introduced a state-dependence to
the AR5-IR carbon cycle. This state-dependent carbon cycle was better able to capture both the observed relationship between
historical emission trajectories and atmospheric $CO_2$ burden; and the behaviour of ESMs in idealised concentration increase
and pulse emission experiments. FaIR v1.0 used four equations to model the atmospheric gas cycle and corresponding effec-
tive radiative forcing (ERF) impact of $CO_2$, and a further two (unchanged from the AR5-IR) to emulate the climate system's
thermal response to changes in ERF. Subsequently, Smith et al. (2018) added a representation of other GHGs and aerosols,
which necessarily increased the structural complexity of the model in FaIR v1.3. In this update, we maintain the ability to
simulate the atmospheric response to a wide range of GHGs and aerosol emissions, while attempting to significantly reduce
the complexity of the model structure.

In FaIRv2.0 we propose a set of six equations that we demonstrate are sufficient to capture the global-mean climate sys-
tem response to GHG and aerosol emissions. These six equations are outlined in figure 1. In this text we explain the physical
reasoning behind each equation and select a default parameter set based on simple tunings to historical observations and recent
literature. We compare the default response of FaIRv2.0 to the a publicly available version of the widely used SCM, MAGICC6
(Meinshausen et al., 2011a, b), for a range of Socioeconomic Pathways (Riahi et al., 2017, SSPs). Further, we show that these
equations can be tuned to emulate key properties of a range of CMIP6 (Eyring et al., 2016) models. Finally, we constrain a large
parameter ensemble inferred from more complex models and contemporary assessments with observations of the present-day
warming level and rate to provide a set of observationally constrained probabilistic projections for the future climate following
(Smith et al., 2018).

FaIRv2.0 is sufficiently simple as to be able to be used in undergraduate and high-school teaching of climate change, and
can illustrate some key properties of the climate system such as the warming impacts of different GHGs, the implications of
uncertainty in ECS and TCR, or the importance of carbon cycle feedbacks. To allow students and other users unfamiliar with
scientific programming languages (such as FaIRv2.0's native language, Python) access to the model, we also provide a version
of FaIRv2.0 written in Excel. We hope that this may open exploration of the climate system to a large group of potential users
who do not have the expertise to run presently-available SCMs. The simplicity of FaIRv2.0 additionally means that although
we provide code in a central, open-source repository, which we strongly recommend is used for most cases, users are not forced
to rely on this. In fact we expect it would be relatively quick to re-create in whatever language users are familiar with, and in
whatever format fits their intended usage.





Here we suggest that the major value of SCMs is in their ability to emulate more complex models, such as has been done in Meinshausen et al. (2011b); Tsutsui (2017, 2020); and in their ability to efficiently integrate massive parameter ensembles for probabilistic climate projection as in Smith et al. (2018); Goodwin et al. (2019). While default parameters must be provided to enable unfamilliar users access to the model, the response arising from these parameters is a function of how they themselves have been selected, rather than one of the model equations themselves. So long as the underlying model equations are

sufficiently flexible to emulate a wide range of climate system responses to the variables of interest (for instance the inferred range of responses within the CMIP ensemble), and have a basis in known physical processes, the SCM should be considered to be valid. Although understanding why the default response of SCMs differ is important, comparisons of solely the default response as a test of how "good" a model is are unhelpful; it is likely that any SCM could be re-tuned to better perform against whatever (single) metric is being used for evaluation, whether another SCM, a more complex model, or something else.


In this study we first (section 2) outline the history and reasoning behind the model equations used, including how we selected default parameters, stepping through the concentration response to emissions; the concentration-forcing relationships; and the thermal response to forcing. We then demonstrate how several key components of FaIRv2.0 – the carbon cycle, aerosol response and thermal response to forcing – can be tuned to emulate a set of CMIP6 models in section 3. section 4 describes the

use of FaIRv2.0 to constrain climate sensitivities and future surface temperature projections using a large ensemble, following Smith et al. (2018). We then provide a discussion of previous comparisons of SCMs in section 5, and suggest some ways in which FaIRv2.0 could be used in section 6 before concluding.





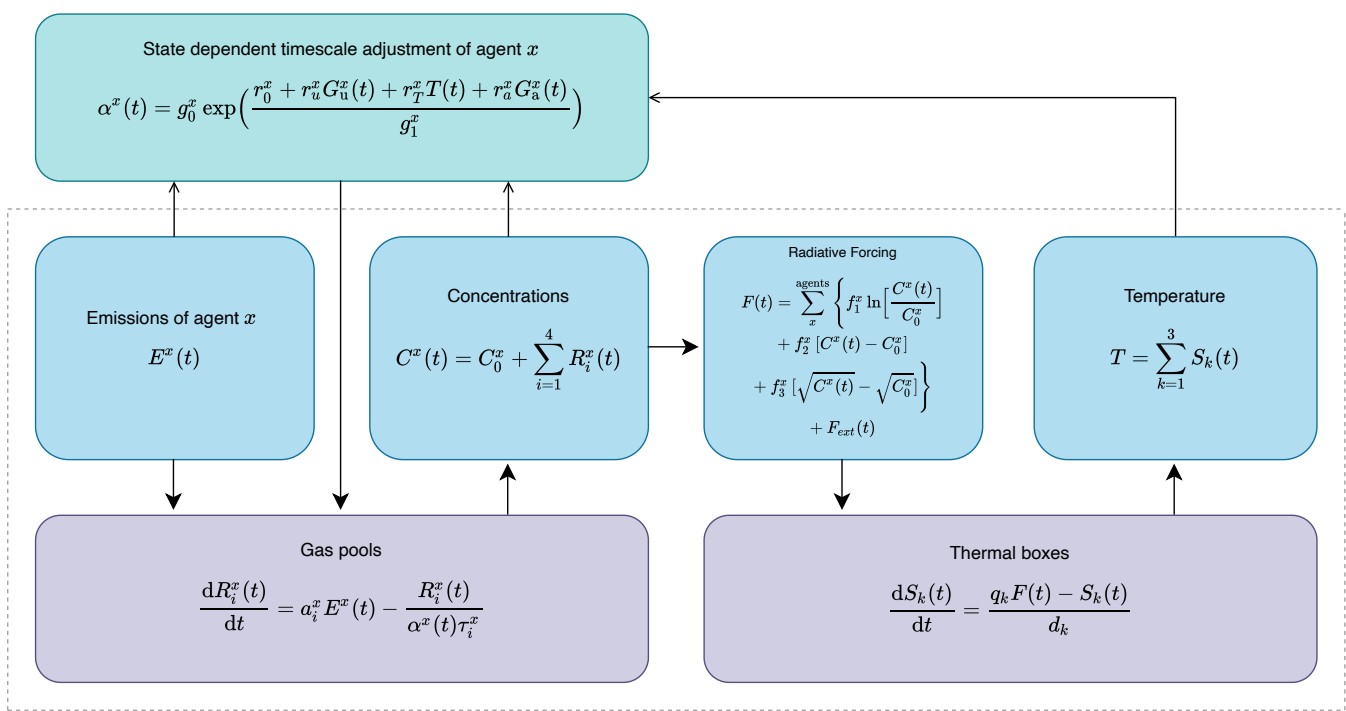

**Figure 1.** Schematic showing the full model structure and equations used. Model steps take place from left to right, with thick arrows indicating the flow of model steps that occur during timestep $t$, and thin arrows indicating steps that occur in between timesteps $t$ and $t + dt$. Equations are described in full below. The dashed grey line indicates the components identical to AR5-IR (Myhre et al., 2013). Table 1 provides brief descriptions of each named parameter in the figure.





**Table 1.** Qualitative analogies for named parameters in FaIRv2.0

| parameter | units | qualitative description |
|---|---|---|
| $E(t)$ | see table S1 | Quantity of agent emitted into atmosphere |
| $C(t)$ | see table S1 | Concentration of agent in atmosphere |
| $C_0$ | unit($C$) | Pre-industrial concentration of agent in atmosphere |
| $R_i(t)$ | unit($E$) | Quantity of agent in $i^{\text{th}}$ atmospheric pool |
| $a_i$ | - | Fraction of emissions entering $i^{\text{th}}$ atmospheric pool |
| $\tau_i$ | yrs | Atmospheric lifetime of gas in $i^{\text{th}}$ pool |
| $\alpha(t)$ | - | Multiplicative adjustment coefficient of pool lifetimes |
| $r_0$ | - | Strength of pre-industrial uptake from atmospheric |
| $r_u$ | unit($E$)$^{-1}$ | Sensitivity of uptake from atmosphere to cumulative uptake of agent since model initialisation |
| $r_T$ | K$^{-1}$ | Sensitivity of uptake from atmosphere to model temperature change since initialisation |
| $r_a$ | unit($E$)$^{-1}$ | Sensitivity of uptake from atmosphere to current atmospheric burden of agent |
| $G_u(t)$ | unit($E$) | Cumulative uptake of agent since model initialisation |
| $T$ | K | Model temperature change since initialisation |
| $G_a(t)$ | unit($E$) | Atmospheric burden of agent above pre-industrial levels |
| $F(t)$ | W m$^{-2}$ | Effective radiative forcing change since the pre-industrial period |
| $f_1$ | W m$^{-2}$ | Logarithmic concentration–forcing coefficient |
| $f_2$ | W m$^{-2}$ unit($C$)$^{-1}$ | Linear concentration–forcing coefficient |
| $f_3$ | W m$^{-2}$ unit($C$)$^{\frac{-1}{2}}$ | Square root concentration–forcing coefficient |
| $S_k(t)$ | K | Response of $k^{\text{th}}$ thermal box |
| $q_k$ | K W$^{-1}$ m$^2$ | Equilibrium response of $k^{\text{th}}$ thermal box |
| $d_k$ | yrs | Response timescale of $k^{\text{th}}$ thermal box |
| $T(t)$ | K | Surface temperature response since model initialisation |

## 2 FaIRv2.0 model framework

As with the previous iteration, FaIRv2.0 is a 0D model of globally averaged variables. It models the GHG emission → concentration → effective radiative forcing (ERF), aerosol emission → ERF, and ERF → temperature responses of the climate system. Here we present the equations behind these responses, separating out the model into the key components.

### 2.1 The gas cycle

FaIRv2.0 inherits the GHG gas cycle equations directly from the carbon cycle equations within FaIRv1.5 (Smith et al., 2018) and v1.0 (Millar et al., 2017). This carbon cycle adapts the 4 pool impulse-response function model in Joos et al. (2013) by introducing a state-dependent timescale adjustment factor, $\alpha$. This factor scales the decay timescale of atmospheric carbon into





each of the 4 pools, allowing for the effective carbon sink from the atmosphere to change in strength. This allows FaIRv2.0 to represent non-linearities in the carbon-cycle in a manner similar to JOOS et al. (1996) or Hooss et al. (2001). In Millar et al. (2017), $\alpha$ was calculated through a parameterisation of the 100-year integrated Impulse Response Function (iIRF$_{100}$, the average airborne fraction over a period of 100 years). In Millar et al. (2017), the iIRF$_{100}$ was parameterised by a simple linear relationship with the quantity of carbon removed since initialisation $G_u$, and the current temperature $T$:

$$\text{iIRF}_{100} = r_0 + r_u G_u + r_T T,$$

where $r_0$ is the initial (pre-industrial) iIRF$_{100}$, and $r_u$ and $r_T$ control how the iIRF$_{100}$ changes as the cumulative carbon uptake and temperature increases. This parameterisation was informed by the behaviour of ESMs and remains consistent with the key feedbacks involved in the carbon cycle (Arora et al., 2019). However, in Millar et al. (2017), the root of a non-linear equation had to be found to update $\alpha$ at each model timestep. The solution of this equation is approximately exponential in iIRF$_{100}$ to a high degree of accuracy for a wide range of values and so in FaIRv2.0, $\alpha$ is calculated using the exponential form given below.

We parameterise this carbon cycle to enable it to simulate a wide range of GHGs, as discussed below. The equations for the carbon cycle and all other gas cycles are, in their most general form:

$$\frac{\mathrm{d}R_i(t)}{\mathrm{d}t} = a_i E(t) - \frac{R_i(t)}{\alpha \tau_i}, \tag{1}$$

$$C(t) = C_0 + \sum_{i=1}^{n} R_i(t) \text{ and} \tag{2}$$

$$\alpha(t) = g_0 \cdot \exp\left(\frac{r_0 + r_u G_u(t) + r_T T(t) + r_a G_a(t)}{g_1}\right); \tag{3}$$

$$\text{where } g_1 = \sum_{i=1}^{n} a_i \tau_i \left[1 - \left(1 + 100/\tau_i\right)e^{-100/\tau_i}\right]$$

$$\text{and } g_0 = \exp\left(-\frac{\sum_{i=1}^{n} a_i \tau_i [1 - e^{-100/\tau_i}]}{g_1}\right).$$

Equations 1 and 2 describe a gas cycle with an atmospheric burden above the pre-industrial concentration, $C_0$, formed of $n$

pools: each pool corresponds to a different sink from the atmosphere. Each pool, $R_i$, has an uptake fraction $a_i$ and decay timescale $\alpha \tau_i$. At each timestep, the state-dependent adjustment, $\alpha$, is computed and the pool concentrations are updated and aggregated to determine the new atmospheric burden. The new atmospheric concentration is then simply the sum of the burden and the pre-industrial concentration. $\alpha$ provides feedbacks to the gas lifetimes based on the current timestep's levels of accumulated emissions ($G_u$), global temperature ($T$), and atmospheric gas burden ($G_a$). $G_a$ is included to enable FaIRv2.0

to emulate the sensitivity of the CH$_4$ lifetime to its own atmospheric burden, as predicted by atmospheric chemistry and simulated in chemical transport models (CTMs) (Holmes et al., 2013; Prather et al., 2015). We also find that the emulation of the carbon-cycle of a number of CMIP6 models over the 1pctCO$_2$ experiment is significantly improved if $G_a$ is included in the iIRF$_{100}$ parameterisation; see 3.2. $g_0$ and $g_1$ set the value and gradient of our analytic approximation for $\alpha$ equal to the

numerical solution of the Millar et al. (2017) iIRF$_{100}$ parameterisation at $\alpha = 1$, and are therefore determined by the final two
equations above and not independent parameters. In the following section, we discuss how we parameterise the gas cycle to
enable FaIRv2.0 to simulate a wide range of GHGs using these same three equations. Qualitative analogies for each parameter
to aid understanding are given in table 1.

Here we emphasize the advantage of using this common framework to simulate the response to all the different GHG
and aerosol emissions: if a user is able to understand the FaIRv2.0 carbon cycle, then they understand how the model will
respond to emissions of any other GHG or aerosol. This is because the carbon cycle is the most complex parameterisation
of the above equations (it is the only gas that is simulated with more than one atmospheric pool as discussed below). This
structural simplicity makes gaining familiarity with the model far easier than if several different gas cycle formulations were
used for different GHGs.

### 2.1.1 Parameterising the gas cycle for a wide range of GHGs

In this section, we consider how the above equations can be parameterised to represent the gas cycles for many different GHGs.
We also provide default parametersations for each GHG, given in full in table S2.

**Carbon dioxide**

As discussed above, FaIRv2.0 retains the state-dependent formulation (Millar et al., 2017) of the 4-pool impulse-reponse
model from Joos et al. (2013); hence $n = 4$. We retain the same state-dependency as in Millar et al. (2017), so the $r$ parameters
are non-zero with the exception of $r_a$. The multi-model mean $a$ and $\tau$ coefficients from Joos et al. (2013) are used by default.
Default $r_u$ and $r_T$ parameters are taken as the mean of the parameter distributions inferred from CMIP6 models in section 4.2.1.
Following Jenkins et al. (2018), we tune the default $r_0$ parameter such that present-day (2018) cumulative $CO_2$ emissions match
the Global Carbon Project (GCP) estimates (Friedlingstein et al., 2019) when historical concentrations (Meinshausen et al.,
2017) are inverted back to emissions by equations 1, 2 and 3. Here we take the GCP dataset as a best-estimate of observed
emissions, but it is important to note that using a different dataset (such as the RCMIP protocol emissions, which are designed
to match the data used in CMIP6 esm- runs) would result in a different value. The pre-industrial concentration is fixed at 278
ppm.

**Methane**

We parameterise methane using a single atmospheric sink: $n = 1$. Although several individual mechanisms have been identified
for the removal of atmospheric methane – tropospheric OH, tropospheric Cl, stratospheric reactions and soil uptake (Prather
et al., 2012; Holmes et al., 2013) – these can be aggregated into a single effective atmospheric lifetime. Through $r_T$ and $r_a$,
we include the key lifetime feedback dependence on to its own atmospheric burden, and tropospheric air temperature and
water vapour mixing ratio (Holmes et al., 2013). We tune $r_a$ to match the sensitivity of the methane lifetime to its own atmo-

spheric burden at the present-day found by Holmes et al. (2013). $r_T$ is tuned to match the sensitivity of the methane lifetime
to tropospheric air temperature and water vapour at the present-day found by Holmes et al. (2013). Since both tropospheric
air temperature and water vapour are closely related to surface air temperatures (they are often approximated by simple pa-
rameterisations of the surface air temperature, as in Holmes et al. (2013)), including these two sensitivities through a single
surface temperature feedback closely replicates lifetime behaviour if both are included separately. $\tau$ is then set such that the

mean emission rate since 2000 matches current estimates from the RCMIP database (Nicholls et al., 2020) when historical
concentrations (Meinshausen et al., 2017) are inverted by FaIRv2.0; and $r_0$ is set such that $\alpha = 1$ at model initialisation. The
pre-industrial concentration is fixed at 720 ppm.

**Nitrous oxide**

Nitrous oxide is parameterised with a single atmospheric sink, and no lifetime sensitivities: $n = 1$ and $\{r_u, r_T, r_a\} = 0$. Al-

though there is evidence that nitrous oxide has a small sensitivity to its atmospheric burden (Prather et al., 2015), when included
in FaIRv2.0 this made very little difference to nitrous oxide concentrations, even under high emission scenarios. We therefore
do not include this additional complexity. $\tau$ is set to match the best-estimate present-day residence lifetime in Prather et al.
(2015) of 109 years; and $r_0$ is set such that $\alpha = 1$ at model initialisation. The pre-industrial concentration is fixed at 270 ppm.

**Halogenated gases**

All other GHGs are treated as having a single atmospheric lifetime and no feedbacks: $n = 1$ and $\{r_u, r_T, r_a\} = 0$. We take
lifetime estimates from WMO (2018). Pre-industrial concentrations (if non-zero) are set to the 1750 value from Meinshausen
et al. (2017). Inclusion of a temperature-dependent lifetime to represent changes to the Brewer-Dobson circulation (Butchart
and Scaife, 2001), as in the MAGICC SCM (Meinshausen et al., 2011a), would be possible through a non-zero $r_T$ parameter.
We do not include a representation of this effect in our default parametersation due to its small impact on model output and

increase in model complexity.

**Aerosols**

Aerosols have considerably shorter lifetimes than the timescales generally considered by SCMs (Kristiansen et al., 2016).
In FaIRv2.0, as in previous iterations (Smith et al., 2018) and other SCMs (Meinshausen et al., 2011a), they are therefore
converted directly from emissions to radiative forcing. In FaIRv2.0, this can be achieved by setting $n = 1$, $\tau = 1$, and providing

a unit conversion factor of 1 between emissions and "concentrations".

### 2.1.2   Historical and SSP concentration trajectories

Here we compare the default parameterisation gas cycle model in FaIRv2.0 to a previous version, FaIRv1.5 (Smith et al.,
2018), and to MAGICC7.1.0-beta (Meinshausen et al., 2019), highlighting any differences. All three models are run under the
fully emission-driven "esm-allGHG" RCMIP protocol (Nicholls et al., 2020); in the case of FaIRv2.0 we use data from the





Global Carbon Project (Friedlingstein et al., 2019) for the input $CO_2$ emissions. FaIRv2.0 matches trajectories from both its previous iteration and the more comprehensive MAGICC closely for all GHGs. We note some discrepancies in the timeseries for halogenated gases between FaIRv2.0 and MAGICC, possibly due to the incorporation of a state-dependent OH abundance and representation of changes to the Brewer-Dobson circulation which modulate the lifetimes of these gases (Meinshausen et al., 2011a). We note that for these gases we could have matched historical concentrations closer by tuning the lifetimes

to the RCMIP emission and historical concentration timeseries (Nicholls et al., 2020; Meinshausen et al., 2017), but argue that taking the best-estimate lifetimes from WMO (2018) is defensible; it is more transparent and avoids emission source dependent parameters (if a different emission dataset were used, the resulting tuned lifetimes would be different). The lower $CO_2$ concentration projections in FaIRv2.0 compared to FaIRv1.5 are due to weaker temperature and cumulative carbon uptake feedbacks (lower $r_u$ and $r_T$) as inferred from the CMIP6 carbon cycle tunings performed in section 3.2.



**Figure 2.** Comparison of historical and future concentration trajectories over a range of SSPs. Units for all GHGs are ppb with the exception of $CO_2$ which is plotted in ppm. Inset panels for $CO_2$, $CH_4$ and $N_2O$ show the historic period.





**Specification of natural emissions**

In FaIRv2.0 we have chosen to formulate the gas cycle equations in terms of a perturbation above the pre-industrial (natural equilibrium) concentration. By definition, this assumes a time-independent quantity of natural emissions for each gas (which can be derived from the pre-industrial concentration and lifetime of the gas). This differs from Meinshausen et al. (2011a) and Smith et al. (2018), who (when driving the respective models with emissions and with the exception of $CO_2$) require a quantity

of natural emissions to be supplied in addition to any anthropogenic emissions by default (though the models can also be run in a fully emission-driven mode as in figure 2). Over the historical period, these emissions are chosen such that they "close the budget" between total anthropogenic emissions, and observed concentrations (Meinshausen et al., 2011a; Smith et al., 2018). This procedure of balancing the budget over history is analogous to driving the model with concentrations up to the present day, and then switching to driving the model with emissions afterwards. While this methodology has the advantage of ensuring

the model simulates present-day concentrations that match observation exactly, it loses consistency between the way in which the model simulates the past and the future. If care is not taken when running these models, this loss of consistency could lead to discontinuities at the present-day (when the model switches from concentration- to emission-driven). As present-day trends are crucial for the estimation of many policy and scientifically relevant quantities such as TCR, TCRE and remaining carbon budgets (Leach et al., 2018; Tokarska et al., 2020; Jiménez-de-la Cuesta and Mauritsen, 2019), we have chosen to enforce a

consistent model (ie. emission- *or* concentration-driven) over the entire simulation period in FaIRv2.0. We note that replicating this budget closing procedure is possible in FaIRv2.0 by inverting observed concentrations to emissions and then joining these inverse emission timeseries to any future scenarios manually. In this study, FaIRv2.0 is run in emission-driven mode unless stated otherwise.

## 2.2 Effective radiative forcing

FaIRv2.0 uses a simple formula to relate atmospheric gas concentrations to effective radiative forcing. This equation, below, includes logarithmic, square-root, and linear terms; motivated by the concentration-forcing relationships in Myhre et al. (2013) of $CO_2$, $CH_4$ and $N_2O$, and all other well-mixed GHGs respectively. For most agents, the concentration- (or for aerosols, emission-) forcing relationship can be reasonably approximated by one of these terms in isolation, however if there is substantial evidence the relationship deviates significantly from any one term, others are able to be included to provide a more

accurate fit. $F_{ext}$ is the sum of all exogenous forcings supplied. These may include natural forcing agents or forcing due to albedo changes.

$$F(t) = \sum_{x}^{\text{forcing agents}} \left\{ f_1^x \cdot \ln\left[\frac{C^x(t)}{C_0^x}\right] + f_2^x \cdot [C^x(t) - C_0^x] + f_3^x \cdot \left[\sqrt{C^x(t)} - \sqrt{C_0^x}\right] \right\} + F_{ext}. \tag{4}$$





### 2.2.1 Parameterising the forcing equation

**Carbon dixoide, nitrous oxide and methane**

We assume the forcing relationship for carbon dioxide is well approximated by the combination of a logarithmic and square root term (Ramaswamy et al., 2001), $f_2^{CO_2} = 0$; both the methane and nitrous oxide concentration-forcing relationships are approximated by a square root term only, $f_{1,2}^{CH_4, N_2O} = 0$. Although overlaps between the spectral bands of these gases mean more complex function forms including interaction terms represent our current best approximation to the observed relationship from spectral calculation (Etminan et al., 2016), inclusion of these interaction terms significantly increases the structural com-

plexity of the model. These overlap terms are most significant for very high concentrations of these gases, and we find that the more simple relationships used here are sufficiently accurate within the context of the uncertainties associated with such high concentration scenarios. We fit the non-zero $f$ coefficients to the Oslo-line-by-line (OLBL) data from Etminan et al. (2016). Our resulting fits have a maximum absolute error of $0.115 \, \mathrm{W \, m^{-2}}$ when compared to the OLBL data, though this is for the most extreme-high concentration data point; and the associated relative error is 1.1%. Figure S1 provides a complete comparison of

how the fit relationships used here compare to the OLBL data, and to the simple formulae which include interaction terms in Etminan et al. (2016).

**Halogenated GHGs**

Following other simple models (Smith et al., 2018; Meinshausen et al., 2011a), we assume concentrations of halogenated gases are linearly related to their direct effective radiative forcing, $f_{1,3}^x = 0$. The conversion coefficient for each gas is its radiative

efficiency, which we take from WMO (2018).

**Aerosol-radiation interaction**

We follow Smith et al. (2020), parameterising the ERF due to aerosol radiation interaction as a linear function of sulphate, organic carbon and black carbon aerosol emissions:

$$\mathrm{ERFari} = f_2^{SO_2} E^{SO_2} + f_2^{OC} E^{OC} + f_2^{BC} E^{BC}. \tag{5}$$

Default parameters are taken as the mean of parameters tuned to 10 CMIP6 models in (Smith et al., 2020, see section 3.3).

**Aerosol-cloud interaction**

ERF due to aerosol-cloud interactions is parameterised following a modification of Smith et al. (2020), as a logarithmic function of sulphate aerosol emissions, and a linear function of organic carbon and black carbon aerosol emissions:

$$\mathrm{ERFaci} = f_1^{aci} \ln\left(1 + \frac{E_{SO_2}}{C_0^{SO_2}}\right) + f_2^{aci}(E^{OC} + E^{BC}). \tag{6}$$

Here $C_0^{SO_2}$ effectively acts as a shape parameter for the logarithmic term. We fit this functional form to the ERFaci component in 10 CMIP6 models derived by the Approximate Partial Radiative Perturbation method (Zelinka et al., 2014) in section 3.3.





The default value for the OC+BC coefficient is taken as the mean of the CMIP6 fits. For the logarithmic coefficient and shape parameter, which appear to have highly skewed distributions due to some models displaying linear and others displaying logarithmic behaviour against sulphate emissions, we calculate default parameters as the mean of the CMIP6 fits, assuming the underlying distribution is lognormally distributed.


**Tropospheric ozone**

Tropospheric ozone is parameterised following (Stevenson et al., 2013) as a linear function of methane concentrations, and nitrate aerosol, carbon monoxide, and volatile organic compound emissions. For the methane component, default coefficients are taken from Holmes et al. (2013). For the others, default coefficients are found by dividing the attributed forcing in (Myhre et al., 2013) by the 2011 emission rate from the Community Emissions Data System (CEDS) inventory (Hoesly et al., 2018).


**Stratospheric ozone**

Stratospheric ozone ERF is parameterised as a linear function of the concentration of ozone depleting substances (ODSs). We assume that the cumulative ERF of a gas over its lifetime is proportional to its ozone depletion potential (WMO, 2018). The best-estimate AR5 value for stratospheric ozone ERF of -0.05 W m$^{-2}$ and observed concentrations of each ODS (Meinshausen et al., 2017) in 2011 then allows us to calculate default linear coefficients for each gas.


**Stratospheric water vapour**

Stratospheric water vapour is assumed to be a linear function of methane concentrations (Smith et al., 2018) due to its small magnitude. The default coefficient is derived from the indirect forcing components in Holmes et al. (2013): $5.5 \times 10^{-5}$ W m$^{-2}$ ppb$^{-1}$.

**Black carbon on snow**

ERF due to light absorbing particles on snow and ice remains a linear function of black carbon emissions (Smith et al., 2018). In AR5, the best estimate of its associated ERF was 0.04 W m$^{-2}$ (Myhre et al., 2013). However, this value is very uncertain, and the efficacy of black carbon on snow may at least double this value (Bond et al., 2013). We therefore calculate our default forcing efficiency by dividing an adopted value of -0.08 W m$^{-2}$ by the CEDS emission rate: 0.011 W m$^{-2}$ MtBC$^{-1}$.

**Contrails**

Combined ERF due to contrails and contrail-induced cirrus is modelled as a linear function of aviation-sector NO$_x$ emissions. The default coefficient is calculated by dividing the best-estimate present-day contrail ERF (Lee et al., 2020) by the CEDS emission rate: 0.011 W m$^{-2}$ MtNO$_x^{-1}$.





**Albedo shift due to land-use change**

In this study we prescribe ERF due to land-use change externally. However, it could be incorporated in a manner identical to FaIRv1.5 by supplying a time-series of cumulative land-use change $CO_2$ emissions, and scaling linearly by a coefficient of -0.00114 W m$^{-2}$ GtC$^{-1}$ (Smith et al., 2018).

## 2.3   Default parameter metric values for comparison

   Table S3 contains default parameter calculated values for the global warming potential (Lashof and Ahuja, 1990) of each
emission type simulated in FaIRv2.0. These values are intended to aid comparison between FaIRv2.0 and other SCMs and do not represent any new analysis.

## 2.4   Temperature response

   The final component of the model calculates the surface temperature response to the changes in ERF. A common representation of this physical process is the energy balance model outlined by Geoffroy et al. (2013). Here we consider the three-box energy
balance model, including the ocean heat uptake efficacy factor introduced by Held et al. (2010). Recent literature has suggested that a two-box energy balance model is insufficient to capture the full range of behaviour observed in CMIP6 models (Tsutsui, 2020, 2017; Cummins et al., 2020). The three-box model can be written in state-space form as:

$$\dot{\mathbf{X}} = A\mathbf{X}, \tag{7}$$

$$\mathbf{Y} = C_d\mathbf{X};$$

where $\quad \mathbf{X} = \begin{pmatrix} F \\ T_1 \\ T_2 \\ T_3 \end{pmatrix}$,

$$A = \begin{pmatrix} 0 & 0 & 0 & 0 \\ 1/C_1 & -(\lambda + \kappa_2)/C_1 & \kappa_2/C_1 & 0 \\ 0 & \kappa_2/C_2 & -(\kappa_2 + \epsilon\kappa_3)/C_2 & \epsilon\kappa_3/C_2 \\ 0 & 0 & \kappa_3/C_3 & -\kappa_3/C_3 \end{pmatrix},$$

$$\mathbf{Y} = \begin{pmatrix} T_1 \\ N \end{pmatrix},$$

and $\quad C_d = \begin{pmatrix} 0 & 1 & 0 & 0 \\ 1 & -\lambda & (1-\epsilon)\kappa_3 & -(1-\epsilon)\kappa_3 \end{pmatrix}.$

   Here, each box $i$ has a temperature $T_i$ and heat capacity $C_i$. $F$ is the prescribed radiative forcing and $N$ is the observed top-
of-atmosphere energy imbalance. Heat exchange coefficients $\kappa$ represent the strength of thermal coupling between boxes $i$ and



$i - 1$. $\lambda$ is the so-called climate feedback parameter. $\epsilon$ is the efficacy factor that enables the energy balance model to account for the variations in $\lambda$ during periods of transient warming observed in GCMs. $T_1$ represents the surface temperature change relative to a pre-industrial climate. For many users of SCMs, the key variable of interest is $T_1$, the surface temperature response. Calculating the surface temperature response can be simplified by diagonalising equation 7, resulting in an impulse-response
in $T_1$ (henceforth referred to as $T$), giving the thermal response form in Millar et al. (2017) (Tsutsui, 2017):

$$\frac{\mathrm{d}S_j(t)}{\mathrm{d}t} = \frac{q_j F(t) - S_j(t)}{d_j} \tag{8}$$

$$\text{and } T(t) = \sum_{j=1}^{3} S_j(t). \tag{9}$$

The response timescales, $d_i$, are given by $\frac{-1}{e_i}$, where $e_i$ are the eigenvalues of $A_{ij;\,2\leq i,j\leq 4}$, and the response coefficients, $q_i$, are given by the product of $e_i$ with the first element of the associated right and left eigenvectors of $A_{ij;\,2\leq i,j\leq 4}$. In FaIRv2.0,
we use this three timescale impulse response form due to its simplicity and flexibility. Two common measures of the climate sensitivity, the equilibrium climate sensitivity (ECS) and transient climate response (TCR) (Collins et al., 2013) are easily expressed in terms of the impulse response parameters:

$$\text{ECS} = F_{2\times\text{CO}_2} \cdot \sum_{i=1}^{3} q_i \tag{10}$$

$$\text{TCR} = F_{2\times\text{CO}_2} \cdot \sum_{i=1}^{3} \left\{ q_i \left( 1 - \frac{d_i}{70} \left[ 1 - e^{-\frac{70}{d_i}} \right] \right) \right\}. \tag{11}$$

The default thermal response parameters in FaIRv2.0 are derived as follows. $d_1$, $d_2$, $d_3$ and $q_1$ are taken as the central value of the CMIP6 inferred distributions described in section 4.2.3. $q_2$ and $q_3$ are then set by equations 10 and 11 such that the default parameter set response has climate sensitivites (ECS and TCR) equal to the central values of the constrained ensemble described in section 4: ECS = 3.2 K and TCR = 1.8 K.

## 3   Emulating complex climate models

In this section we demonstrate the ability of FaIRv2.0 to emulate the more complex models from CMIP6 (Eyring et al., 2016) in a limited set of experiments. Due to constraints on data availability, we have focussed on tuning the key components of the model: the carbon cycle; the thermal response; and the aerosol ERF relationships. We use the abrupt-4xCO2 and 1pctCO2 CMIP6 experiments to tune the carbon cycle and thermal response. The highly idealised nature of these experiments means that parameters arising from these tunings will not necessarily be able to emulate complex model response to more realistic
scenarios due to processes that FaIRv2.0 cannot represent. In the near future we hope to be able to tune to the historical and SSP CMIP6 experiments in order to validate the tunings given here.





## 3.1 Tuning the thermal response

We follow the statistically rigorous methodology of Cummins et al. (2020) to tune thermal response parameters to 40 CMIP6 models. This involves fitting parameters to the energy balance model above by recursively computing the likelihood via the Kalman filter; the optimal parameters are those that maximise the computed likelihood. We then transform the calculated optimal energy balance parameters into the impulse response form used in FaIRv2.0. We obtain model data from the "abrupt-4xCO2", "1pctCO2" and "piControl" experiments for the top-of-energy imbalance and surface temperature response from ESGF (Cinquini et al., 2014). We calculate anomalies and correct for model drift in the abrupt-4xCO2 and 1pctCO2 experiments using output from the piControl experiment, based on the CMOR parent branch time metadata. To reduce noise in the input data, we average over all available ensemble members for each model. For full details of this anomaly-correction procedure, see the supplement. The Cummins et al. (2020) methodology uses surface temperatures and top-of-atmosphere energy imbalances from the abrupt-4xCO2 experiment to return all the parameters of the energy balance model above, plus the radiative forcing arising from the quadrupling of carbon dioxide concentrations. While this would fully specify both the thermal response and the concentration-forcing relationship if concentration-forcing was a pure logarithmic relationship, several models display significant deviations from a pure logarithmic concentration-forcing relationship (Tsutsui, 2020, 2017). We account for this within the FaIR framework by assuming that the concentration-forcing relationship can be reasonably approximated by the sum of a logarithmic and square-root term. Best-estimate $f_1^{CO_2}$ and $f_3^{CO_2}$ parameters are found by first deriving the TCR of each model using the 1pctCO2 experiment. We can use the tuned impulse-response parameters and TCR to then calculate the forcing at a doubling of carbon dioxide using the relationship above. The forcings at carbon dioxide doubling and quadrupling uniquely specify $f_1^{CO_2}$ and $f_3^{CO_2}$ values for use in FaIR. The best-estimate impulse-response and forcings at carbon dioxide doubling and quadrupling are given below. Corresponding energy balance model parameters are given in the supplement. Figure 3 shows the emulated and original responses to the abrupt-4xCO2 and 1pctCO2 experiments for each model.

**Table 2.** Tuned CMIP6 thermal response parameters



| model | d1 | d2 | d3 / yrs | q1 | q2 | q3 / K W$^{-1}$ m$^2$ | F_2x | F_4x / W m$^{-2}$ | ECS | TCR / K |
|---|---|---|---|---|---|---|---|---|---|---|
| ACCESS-CM2 | 0.8140 | 8.980 | 339.0 | 0.168000 | 0.508 | 0.810000 | 3.18 | 7.20 | 4.72 | 2.18 |
| ACCESS-ESM1-5 | 0.6380 | 6.010 | 342.0 | 0.119000 | 0.447 | 0.865000 | 3.53 | 6.55 | 5.05 | 2.15 |
| AWI-CM-1-1-MR | 1.1600 | 6.950 | 170.0 | 0.222000 | 0.296 | 0.337000 | 3.96 | 7.85 | 3.39 | 2.16 |
| BCC-CSM2-MR | 1.2200 | 7.230 | 225.0 | 0.176000 | 0.341 | 0.566000 | 3.28 | 6.16 | 3.55 | 1.83 |
| BCC-ESM1 | 2.0100 | 9.880 | 282.0 | 0.289000 | 0.305 | 0.539000 | 3.18 | 6.25 | 3.60 | 1.92 |
| CAMS-CSM1-0 | 0.3360 | 4.230 | 132.0 | 0.072500 | 0.297 | 0.166000 | 4.61 | 8.74 | 2.47 | 1.79 |
| CESM2 | 4.1300 | 81.700 | 929.0 | 0.676000 | 0.644 | 1.120000 | 2.58 | 5.52 | 6.28 | 2.28 |
| CESM2-FV2 | 0.5670 | 4.600 | 427.0 | 0.093900 | 0.448 | 1.280000 | 3.35 | 7.39 | 6.09 | 2.04 |
| CESM2-WACCM | 0.3180 | 4.740 | 330.0 | 0.052700 | 0.470 | 0.872000 | 3.71 | 8.06 | 5.17 | 2.14 |
| CESM2-WACCM-FV2 | 0.6350 | 6.220 | 469.0 | 0.140000 | 0.461 | 1.190000 | 2.99 | 6.74 | 5.34 | 1.92 |
| CIESM | 1.1600 | 7.600 | 242.0 | 0.152000 | 0.424 | 0.867000 | 3.91 | 8.38 | 5.64 | 2.51 |
| CNRM-CM6-1 | 1.5900 | 25.700 | 1160.0 | 0.352000 | 0.401 | 0.045400 | 3.25 | 8.74 | 2.59 | 1.98 |
| CNRM-CM6-1-HR | 1.1700 | 11.600 | 233.0 | 0.235000 | 0.454 | 0.293000 | 3.92 | 7.94 | 3.85 | 2.55 |
| CNRM-ESM2-1 | 1.7300 | 11.200 | 367.0 | 0.280000 | 0.542 | 0.683000 | 2.59 | 6.01 | 3.90 | 2.04 |
| CanESM5 | 1.1600 | 11.300 | 296.0 | 0.232000 | 0.571 | 0.770000 | 3.43 | 7.42 | 5.39 | 2.71 |
| E3SM-1-0 | 0.9690 | 11.300 | 275.0 | 0.203000 | 0.698 | 0.832000 | 3.52 | 7.00 | 6.10 | 3.10 |
| EC-Earth3-Veg | 0.8550 | 7.570 | 119.0 | 0.201000 | 0.400 | 0.587000 | 3.59 | 7.47 | 4.27 | 2.51 |
| GFDL-CM4 | 0.0298 | 2.370 | 253.0 | 0.000024 | 0.427 | 0.558000 | 4.20 | 8.95 | 4.14 | 2.03 |
| GFDL-ESM4 | 1.1300 | 7.890 | 292.0 | 0.227000 | 0.253 | 0.154000 | 3.39 | 7.87 | 2.15 | 1.58 |
| GISS-E2-1-G | 0.9460 | 5.610 | 369.0 | 0.202000 | 0.226 | 0.233000 | 4.28 | 8.14 | 2.83 | 1.83 |
| GISS-E2-1-H | 1.5400 | 35.500 | 1.11e8 | 0.318000 | 0.278 | 0.000075 | 4.61 | 8.38 | 2.75 | 2.15 |
| GISS-E2-2-G | 0.8250 | 9.900 | 896.0 | 0.209000 | 0.235 | 0.038500 | 4.04 | 8.20 | 1.95 | 1.65 |
| HadGEM3-GC31-LL | 0.8600 | 9.310 | 279.0 | 0.167000 | 0.604 | 0.863000 | 3.30 | 7.22 | 5.39 | 2.60 |
| HadGEM3-GC31-MM | 1.1200 | 12.300 | 237.0 | 0.276000 | 0.477 | 0.762000 | 3.36 | 7.20 | 5.10 | 2.58 |
| INM-CM4-8 | 1.0700 | 6.190 | 79.3 | 0.207000 | 0.224 | 0.201000 | 2.83 | 5.93 | 1.79 | 1.35 |
| INM-CM5-0 | 1.1300 | 7.330 | 165.0 | 0.217000 | 0.235 | 0.185000 | 2.92 | 6.29 | 1.86 | 1.33 |
| IPSL-CM6A-LR | 1.0600 | 13.500 | 366.0 | 0.316000 | 0.504 | 0.634000 | 3.06 | 6.98 | 4.46 | 2.38 |
| KACE-1-0-G | 0.0318 | 6.260 | 345.0 | 0.029900 | 0.480 | 0.867000 | 3.68 | 7.11 | 5.07 | 2.02 |
| MIROC-ES2L | 3.5700 | 19.400 | 552.0 | 0.324000 | 0.168 | 0.113000 | 3.84 | 7.89 | 2.33 | 1.68 |
| MIROC6 | 1.1000 | 8.470 | 440.0 | 0.268000 | 0.178 | 0.246000 | 3.56 | 7.80 | 2.46 | 1.56 |
| MPI-ESM1-2-HR | 1.7200 | 9.630 | 254.0 | 0.298000 | 0.167 | 0.378000 | 3.52 | 7.82 | 2.97 | 1.70 |
| MPI-ESM1-2-LR | 2.4600 | 72.700 | 6.72e6 | 0.375000 | 0.172 | 0.032000 | 4.19 | 9.48 | 2.43 | 1.77 |
| MRI-ESM2-0 | 1.1100 | 5.280 | 247.0 | 0.163000 | 0.286 | 0.444000 | 3.48 | 7.47 | 3.11 | 1.68 |
| NESM3 | 1.2400 | 22.900 | 445.0 | 0.472000 | 0.345 | 0.267000 | 3.75 | 7.86 | 4.06 | 2.70 |
| NorCPM1 | 2.5000 | 42.500 | 1.77e8 | 0.334000 | 0.246 | 0.065500 | 3.60 | 7.82 | 2.33 | 1.61 |
| NorESM2-LM | 0.2310 | 0.938 | 1350.0 | 0.000073 | 0.250 | 1.080000 | 5.47 | 11.70 | 7.30 | 1.50 |
| NorESM2-MM | 0.4290 | 1.210 | 302.0 | 0.000088 | 0.292 | 0.217000 | 4.19 | 10.80 | 2.13 | 1.30 |
| SAM0-UNICON | 0.8180 | 4.580 | 308.0 | 0.105000 | 0.405 | 0.459000 | 4.57 | 8.33 | 4.43 | 2.42 |
| TaiESM1 | 1.1600 | 6.750 | 274.0 | 0.152000 | 0.428 | 0.553000 | 4.07 | 8.15 | 4.61 | 2.45 |
| UKESM1-0-LL | 0.7530 | 10.200 | 277.0 | 0.210000 | 0.552 | 0.769000 | 3.60 | 7.38 | 5.51 | 2.76 |



**Figure 3.** FaIRv2.0 emulation of CMIP6 model response to the abrupt4xCO2 and 1pctCO2 experiments. Small orange and blue dots show drift-corrected model surface temperature anomaly output for the abrupt-4xCO2 and 1pctCO2 experiments respectively. Dashed black lines show corresponding FaIRv2.0 emulation. Large orange and blue dots over the y-axis indicate the assessed model ECS and TCR respectively (see table 2).





We found that optimal parameters found over the first 150 years of the abrupt-4xCO2 experiment were not able to well-
reproduce the remainder of the experiment for those models that continued the experiment past 150 years (CESM2 and
NorESM2-LM). These models appear to exhibit particularly high ocean heat uptake efficacy, resulting in a sharp "elbow"
feature in their Gregory plots (see figure S2), which tends to be underestimated by the maximum likelihood procedure when
tuning to abbreviated data. Without longer runs from more models, it is difficult to predict whether the projection issues with
tuning parameters to the first 150 years observed in these two models would apply more generally. Rugenstein et al. (2020)
suggests that the inclusion of an ocean heat uptake efficacy in the fit should alleviate this issue to a limited extent.

## 3.2  Tuning the carbon cycle response

We tune the carbon cycle using CMIP6 data from the C4MIP (Jones et al., 2016) fully coupled and biogeochemically coupled
1pctCO2 runs (Arora et al., 2019). Since constraining the response coefficients $a_i$ and timescales $\tau_i$ requires pulse-emission
experiments such as carried out by Joos et al. (2013), here we only fit the $r$ feedback parameters and keep the response
coefficients, $a$, and timescales, $\tau$, equal to the multi-model mean from Joos et al. (2013). The inclusion of both the fully coupled
and biogeochemically coupled runs in the procedure allows us to constrain $r_u$, $r_a$, and $r_T$ independently. We use equations 1
and 2 to diagnose the values of $\alpha$ required to reproduce the C4MIP emissions from the corresponding concentrations within the
FaIRv2.0 carbon-cycle impulse-response framework. We then use equation 3 to convert $\alpha$ into iIRF$_{100}$ timeseries. Finally, we
use an ordinary least squares estimator to calculate $r$ parameters by regressing the C4MIP cumulative uptake, temperature and
atmospheric burden timeseries against the diagnosed iIRF$_{100}$ timeseries. $r_0$ is taken as the intercept of the estimator. We include
the atmospheric burden as a predictor (and hence obtain non-zero $r_A$ values) due to a significant reduction in regression residual
for several models when included. We find that all the C4MIP models display an exceptionally high, rapidly decreasing initial
airborne fraction. In terms of the FaIRv2.0 equations, this corresponds to an $\alpha$ value that decreases initially before reaching a
minimum, representing a carbon sink that initially increases in strength when concentrations start to rise before decreasing as
the concentrations and temperatures rise further. FaIRv2.0 is unable to fully capture this initial adjustment, and as such in our
tunings we prioritise emulating the long-term behaviour and carry out the regression over the final 75 years of the C4MIP data.
It would be possible to better capture the initial adjustment by including additional terms in equation 3, but since it remains
to be seen whether this behaviour is apparent in scenarios where concentrations do not rise suddenly and rapidly from a pre-
industrial level as is the case in the 1pctCO$_2$ experiment (such as a historical emission scenario), we do not do so here. Tuned
parameters are given below, with figure 4 showing diagnosed C4MIP emissions and the FaIRv2.0 emulation. We note that
these tunings suggest that the pre-industrial sink strength (which is encapsulated by $r_0$) in all but one of the models is higher
than the historically observed best-estimate found here (2.1.1), and in a previous study (Jenkins et al., 2018).

**Table 3.** Tuned CMIP6 carbon-cycle parameters





| | $r_0$ | $r_u$ | $r_T$ | $r_a$ |
|---|---|---|---|---|
| ACCESS-ESM1-5 | 32.8 | 0.048200 | 3.4400 | -0.00627 |
| BCC-CSM2-MR | 27.7 | 0.001590 | 4.5200 | 0.00873 |
| CESM2 | 41.3 | 0.007510 | 1.1900 | 0.00626 |
| CNRM-ESM2-1 | 38.5 | -0.000362 | 2.4400 | 0.01030 |
| CanESM5 | 35.9 | -0.007120 | -0.0847 | 0.01910 |
| GFDL-ESM4 | 35.8 | 0.017700 | 4.5100 | -0.00156 |
| IPSL-CM6A-LR | 34.8 | 0.009360 | 0.9340 | 0.01560 |
| MIROC-ES2L | 34.3 | 0.010400 | 3.2900 | 0.00574 |
| MPI-ESM1-2-LR | 35.7 | 0.020800 | 1.2600 | 0.00357 |
| NorESM2-LM | 41.3 | 0.005590 | 1.4500 | 0.00757 |
| UKESM1-0-LL | 38.5 | 0.018100 | 2.5800 | 0.00287 |



**Figure 4.** FaIRv2.0 emulation of CMIP6 model carbon cycle response to the C4MIP 1pctCO2 experiments. Orange and blue dots show model annual and cumulative emissions for the experiments specified above the figure. Dashed black lines show corresponding FaIRv2.0 emulation. Red dots in the right-hand column show cumulative emissions arising from radiation feedback directly from the radiatively-coupled 1pctCO2 experiment, while blue dots show cumulative emissions calculated as the difference between the equivalent fully- and biogeochemically-coupled experiments.





### 3.3 Tuning aerosol ERF

Aerosol forcing relationships are tuned to ERF data from 10 CMIP6 models and emission data from the RCMIP protocol (Nicholls et al., 2020) following Smith et al. (2020). For each CMIP6 model, aerosol-radiation and aerosol-cloud interaction components of the ERF are calculated by the Approximate Partial Radiative Perturbation (APRP) method. For additional details on the exact procedure, see Smith et al. (2020) and Zelinka et al. (2014). For each model, we fit the $f$ coefficients in equation 5 to the ERFari component using an ordinary least squares estimator. The resulting coefficients are almost identical to those from Smith et al. (2020), with differences arising only due to the emission data used. We then fit the $f$ coefficients and $C_0^{SO_2}$ in equation 6 to the ERFaci component by minimising the residual sum of squares using a simplex algorithm (Nelder and Mead, 1965). The tuned parameters are given below. Figure 5, following figure 2 of Smith et al. (2020), shows the parameterised fits compared to the APRP derived model ERF components.

**Table 4.** Tuned CMIP6 aerosol forcing parameters

| Model | ERFari | | | ERFaci | | |
|---|---|---|---|---|---|---|
| | $f_2^{BC}$ | $f_2^{OC}$ | $f_2^{SO_2}$ | $f_1^{aci}$ | $C_0^{SO_2}$ | $f_2^{aci}$ |
| CanESM5 | 0.03260 | -0.000347 | -0.002490 | -0.387 | 23.8 | -0.015200 |
| E3SM | 0.02480 | -0.012600 | -0.000942 | -1.640 | 113.0 | -0.014200 |
| GFDL-CM4 | 0.02690 | -0.002090 | -0.002610 | -2.230 | 427.0 | -0.008030 |
| GFDL-ESM4 | 0.10200 | -0.030400 | -0.002640 | -57.600 | 17000.0 | -0.015300 |
| GISS-E2-1-G | 0.14600 | -0.044100 | -0.006680 | -0.156 | 16.8 | -0.017600 |
| HadGEM3-GC31-LL | 0.00196 | 0.004150 | -0.002910 | -0.783 | 66.9 | -0.006910 |
| IPSL-CM6A-LR | -0.05610 | 0.008850 | -0.000748 | -0.951 | 306.0 | -0.001730 |
| MIROC6 | 0.03870 | -0.014200 | -0.001780 | -0.392 | 46.6 | -0.012400 |
| NorESM2-LM | 0.00302 | -0.003400 | -0.001260 | -68.600 | 10300.0 | -0.012300 |
| UKESM1-0-LL | 0.00255 | 0.000063 | -0.002390 | -0.740 | 38.9 | -0.000265 |



**Figure 5.** Top row shows entire CMIP6 model ensemble compared to the FULL and CONSTRAINED FaIRv2.0 parameter ensembles described in 4. All other figures show FaIRv2.0 emulation of individual CMIP6 models. Orange dots show CMIP6 model output derived using APRP. Black dashed lines show FaIRv2.0 emulation. All series displayed are relative to zero effective radiative forcing in 1850.





## 4 Constraining probabilistic parameter ensembles

The computational efficiency of SCMs makes them an ideal tool for carrying out large ensemble simulations from which
probabilistic projections can be derived. Smith et al. (2018) carried out such a large ensemble, and produced projections
based on constraining the ensemble members to fall within the 5-95% uncertainty range in observed warming to date from the
Cowtan and Way dataset (Cowtan and Way, 2014). Here we replicate this procedure with the new model, but with an additional
constraint on the current rate of warming, and updated prior parameters distributions.

### 4.1 The current level and rate of warming

We determine the current level and rate of warming following the Global Warming Index methodology (Haustein et al., 2017).
This takes into account multiple sources of uncertainty: observational, forcing, earth system response (through parameter
variation in an identical thermal response model to the one used in FaIR) and internal variability. We find that a 90% credible
interval on the 2014 level of warming relative to 1861-1880 is 0.75 - 1.26 K; and on the 2010-2014 rate of warming is 0.18
- 0.40 K decade$^{-1}$. We use 2014 as the year in which our constraint is applied as this is the final year for which historical
aerosol emissions – used in the calculation of the forcing timeseries – are available (Hoesly et al., 2018). We could extend the
calculation to the present day using the SSP2-4.5 emission scenario as was done in Smith et al. (2020), but the rapid reduction
in sulfate emissions following 2014 in this scenario projects significantly onto the rate of warming constraint and this decline
does not appear to reflect observed aerosol forcing trends from atmospheric reanalyses (Bellouin et al., 2020). We note that the
interval obtained for the level of warming here is larger than previous estimates (Haustein et al., 2017; Leach et al., 2018); this
is largely due to updates to the aerosol forcing timeseries. For full details of the calculation, see the supplement.

#### 4.1.1 Definition of global mean temperature

Recent studies (Richardson et al., 2016, 2018) have shown that the definition of globally averaged surface temperature used
is important when comparing observations to climate model output, and is relevant when exploring policy-relevant quantities
such as the carbon budget (Tokarska et al., 2019). Discrepancies arise since observations blend air temperatures over land
and sea ice with water temperature over ocean, and do not have full global coverage (they are blended-masked); while climate
model surface temperature output is globally complete, and always measured as the air temperature 2m above the surface of the
Earth. It has been shown both historically, and over future climate scenarios (Richardson et al., 2018), that the blended-masked
temperature definition (GMST) may be cooler than the globally complete 2m air temperature definition (GSAT). In our Global
Warming Index calculation, we use the mean of 5 temperature observation datasets (Lenssen et al., 2019; Cowtan and Way,
2014; Vose et al., 2012; Morice et al., 2011; Rohde et al., 2013) following the IPCC Special Report on 1.5°C warming (IPCC,
2018); this implies that our constrained ensemble will measure surface temperatures using the GMST definition. This will lead
to slightly lower model estimates of surface temperature than if we used the GSAT definition. We can estimate the difference
between our definition of GMST and GSAT by regressing the 5-dataset mean used here against GSAT from ERA5 (Hersbach
et al., 2020). A least squares estimator suggests that our GMST definition is 3.4 $\pm$ 0.01 % smaller than GSAT.





### 4.2 Sampled prior distributions

#### 4.2.1 Carbon cycle parameters

While including the atmospheric burden is necessary to well-emulate the carbon-cycle behaviour of individual C4MIP models, parameterising the iIRF$_{100}$ as a linear function of just cumulative carbon uptake and temperature is sufficient to capture the spread of the model ensemble. Correlations between parameters also complicate sampling from the inferred parameter distributions derived from table 3. We therefore repeat the parameter tuning procedure described in section 3.2, but exclude the atmospheric burden as a predictor for the C4MIP iIRF$_{100}$ timeseries. The resulting $r_0$, $r_u$ and $r_T$ parameter samples are uncorrelated. We sample these parameters by applying scaling factors inferred from the CMIP6 tunings to the default parameter values (for $r_u$ and $r_T$ this is equivalent to sampling directly from the distribution inferred from the CMIP6 tunings). The underlying uncorrelated scaling factor distributions are given below.

**Table 5.** Carbon-cycle parameter sampling

| parameter | default value | scaling factor, $X$ |
|---|---|---|
| $r_0$ | 30.4 | $X \sim \mathcal{N}(1, 0.122)$ |
| $r_u$ | 0.0177 | $\ln(X) \sim \mathcal{N}(0, 0.116)$ |
| $r_T$ | 2.64 | $X \sim \mathcal{N}(1, 0.613)$ |

#### 4.2.2 Forcing parameters

Uncertainty in effective radiative forcing is included by grouping individual forcing agents into broader forcing classes (IPCC et al., 2013), and applying a randomly sampled scaling factor to all the $f$ parameters within each class (with the exception of aerosol forcings, which we discuss below). Scaling factors between forcing classes are uncorrelated. The scaling factor distributions are given below. Uncertainty in aerosol forcing is included as follows. ERFari $f$ coefficients (equation 5) are first drawn from a multivariate normal distribution inferred from the CMIP6 tuned parameters in 4. We then apply a quantile map to scale the resulting coefficients such that the 1850 to 2005-2015 mean ERFari distribution matches the process based assessment in Bellouin et al.. For ERFaci, $f_2^{aci}$ coefficients (equation 6) are drawn from a normal distribution inferred from the CMIP6 tuned parameters in 4. $f_1^{aci}$ and $C_0^{SO_2}$ coefficients are drawn from a multivariate log-normal distribution; this ensures we sample the full range of ERFaci shapes provided by CMIP6 models. As with the ERFari coefficients, we then apply a quantile map to scale these coefficients such that the sampled 1850 to 2005-2015 mean ERFaci distribution matches Bellouin et al.. The underlying scaling factor distributions used for each forcing class (excluding aerosol forcing) are given below.

**Table 6.** ERF parameter sampling



| forcing category | scaling factor, $X$ | 5-95% uncertainty (%) |
|---|---|---|
| $CO_2$ | $X \sim \mathcal{N}(1, 0.122)$ | $\pm 20$ |
| $CH_4$ | $X \sim \mathcal{N}(1, 0.170)$ | $\pm 28$ |
| $N_2O$ | $X \sim \mathcal{N}(1, 0.122)$ | $\pm 20$ |
| other WMGHGs | $X \sim \mathcal{N}(1, 0.122)$ | $\pm 20$ |
| tropospheric ozone | $X \sim \mathcal{N}(1, 0.304)$ | $\pm 50$ |
| stratospheric ozone | $X \sim \mathcal{N}(1, 1.22)$ | $\pm 200$ |
| stratospheric $H_2O$ from $CH_4$ | $X \sim \mathcal{N}(1, 0.438)$ | $\pm 72$ |
| black carbon on snow | $\ln(X) \sim \mathcal{N}(0, 0.457)$ | - |
| contrails | $X \sim \mathcal{N}(1, 0.456)$ | $\pm 75$ |
| land use change | $X \sim \mathcal{N}(1, 0.456)$ | $\pm 75$ |
| volcanic | $X \sim \mathcal{N}(1, 0.304)$ | $\pm 50$ |
| solar | $X \sim \mathcal{N}(1, 0.608)$ | $\pm 100$ |

### 4.2.3 Thermal response parameters

Uncertainty in thermal response is incorporated by sampling response parameters directly from distributions inferred from
the CMIP6 tunings in section 2, taking correlations between parameters into account. Referring to parameters as in equations
8, 9, 10 and 11, we draw parameters from the following distributions. $d_1$, $d_2$ and $q_1$ are highly correlated, and we therefore
sample $\ln(d_1)$, $\ln(d_2)$ and $q_1$ from a multivariate normal distribution with covariances and means taken from the values in
section 2. $d_3$ is not strongly correlated with any other parameter, and so we sample $\ln(d_3)$ from a normal distribution. We then
independently sample the TCR and the TCR/ECS ratio, the Realised Warming Fraction (RWF), as it has been shown that the
TCR and RWF are much more weakly correlated than any other combination of ECS, TCR and RWF (Millar et al., 2015). We
draw TCR samples from a normal distribution, $\text{TCR} \sim \mathcal{N}(2, 0.608)$, truncating the distribution at a distance of $\pm 3\sigma$ from the
central value of 2. We draw RWF samples from a normal distribution $\text{RWF} \sim \mathcal{N}(0.55, 0.15)$, again truncating at $\pm 3\sigma$. The
90% credible interval of the sampled TCR and RWF distributions closely, but not exactly, match the ranges inferred from the
parameters in table 2. Using equations 10 and 11, we then calculate $q_2$ and $q_3$. We reject any samples in which any of the $q$
parameters are unphysical (negative). The quantiles of the underlying ECS and TCR distributions used are given below.

### 4.3 The constrained ensemble

Taking historical $CO_2$ emissions from GCP (Friedlingstein et al., 2019), and all other historical and future SSP (Riahi et al.,
2017) emissions from the RCMIP protocol (Nicholls et al., 2020); and land use change, volcanic, and solar forcing from the
SSP effective radiative forcing timeseries (Smith, 2020), we run a 1,000,000 member emission-driven ensemble (FULL),
sampling uncertainty in the carbon cycle, effective radiative forcing and thermal response as described above. We then
constrain this ensemble (CONSTRAINED) based on the assessed 90% credible intervals of the 2014 level and rate of the
Global Warming Index (Haustein et al., 2017) as above (4.1). Table 7 gives outlines the results of this analysis in terms of the





quantiles of key variables.

**Table 7.** Constrained ensemble results for climate sensitivities and current ERF. ERF in 2019 is based on following an SSP2-4.5 pathway from 2014 onwards.

| | FULL | | | | | CONSTRAINED | | | | |
|---|---|---|---|---|---|---|---|---|---|---|
| climate sensitivity | 5% | 17% | 50% | 83% | 95% | 5% | 17% | 50% | 83% | 95% |
| ECS | 1.91 | 2.55 | 3.73 | 5.43 | 7.26 | 1.94 | 2.36 | 3.17 | 4.45 | 5.84 |
| TCR | 1.18 | 1.53 | 2.06 | 2.62 | 3.02 | 1.25 | 1.45 | 1.77 | 2.13 | 2.41 |
| 2019 ERF relative to 1750 / W m$^{-2}$ | | | | | | | | | | |
| $CO_2$ | 1.62 | 1.80 | 2.08 | 2.37 | 2.58 | 1.67 | 1.84 | 2.11 | 2.38 | 2.59 |
| $CH_4$ | 0.45 | 0.53 | 0.63 | 0.73 | 0.81 | 0.45 | 0.52 | 0.63 | 0.73 | 0.80 |
| $N_2O$ | 0.16 | 0.18 | 0.20 | 0.22 | 0.24 | 0.16 | 0.18 | 0.20 | 0.22 | 0.24 |
| other WMGHGs | 0.29 | 0.32 | 0.36 | 0.40 | 0.43 | 0.29 | 0.32 | 0.36 | 0.40 | 0.43 |
| tropospheric $O_3$ | 0.20 | 0.28 | 0.40 | 0.51 | 0.59 | 0.20 | 0.28 | 0.39 | 0.51 | 0.59 |
| stratospheric $O_3$ | -0.14 | -0.10 | -0.05 | 0.01 | 0.05 | -0.14 | -0.10 | -0.05 | 0.01 | 0.05 |
| stratospheric $H_2O$ from $CH_4$ | 0.02 | 0.04 | 0.06 | 0.09 | 0.11 | 0.02 | 0.04 | 0.06 | 0.09 | 0.11 |
| total WMGHGs | 3.14 | 3.36 | 3.69 | 4.02 | 4.27 | 3.18 | 3.39 | 3.71 | 4.03 | 4.27 |
| ERFari | -0.59 | -0.46 | -0.30 | -0.14 | -0.02 | -0.58 | -0.46 | -0.31 | -0.16 | -0.05 |
| ERFaci | -2.26 | -1.46 | -0.69 | -0.26 | -0.05 | -1.25 | -1.00 | -0.63 | -0.30 | -0.11 |
| total aerosols | -2.59 | -1.79 | -1.00 | -0.52 | -0.26 | -1.56 | -1.32 | -0.95 | -0.60 | -0.37 |
| black carbon on snow | 0.04 | 0.05 | 0.09 | 0.13 | 0.18 | 0.04 | 0.05 | 0.08 | 0.13 | 0.18 |
| contrails | 0.01 | 0.02 | 0.04 | 0.06 | 0.07 | 0.01 | 0.02 | 0.04 | 0.06 | 0.07 |
| total anthropogenic | 0.90 | 1.73 | 2.59 | 3.22 | 3.63 | 1.98 | 2.25 | 2.67 | 3.13 | 3.46 |

### 4.3.1 Current effective radiative forcing

The constraint applied only significantly affects the estimated ranges of ERFaci, total aerosol and anthropogenic forcings in
2019 (based on an SSP2-45 pathway following 2014). ERFaci is constrained from -0.69 [-2.26 , -0.05][1] to -0.63 [-1.25 , -0.11];
total aerosol forcing from -1.00 [-2.59 , -0.26] to -0.95 [-1.55 , -0.37]; and total anthropogenic forcing from 2.59 [0.90 , 3.63] to
2.67 [1.98 , 3.46]. These results are consistent with a recent study that used similar methods but concentrated on aerosol forcing
and used a constraint based on observed warming and Earth energy uptake (Smith et al., 2020). Other forcing categories are
not affected by the constraint due to their relatively smaller magnitude and/or prior uncertainty.

---

[1] square brackets indicate a 90% credible interval





### 4.3.2 Climate sensitivities

We find that the TCR is constrained from 2.06 [1.18 , 3.02] to 1.77 [1.25 , 2.41] and the ECS from 3.73 [1.91 , 7.26] to 3.17 [1.94 , 5.84]. These results are relatively consistent with several recent studies that have used emergent constraint techniques (Nijsse et al., 2020; Jiménez-de-la Cuesta and Mauritsen, 2019; Tokarska et al., 2020) or drawn on multiple lines of evidence (Sherwood et al., 2020). The largest discrepancies with these studies occur at the upper tails of the constrained distributions; the constraint applied here is unable to rule out some higher values of the ECS which these other studies have done, though we note that the 95[th] percentile of the CONSTRAINED ECS distribution is only slightly above the robust upper bound in Sherwood et al. (2020). The CONSTRAINED RWF distribution does not differ significantly from the FULL distribution of 0.55 [0.3 , 0.8].

**Figure 6.** Climate sensitivities of our FULL, CONSTRAINED and ALTERNATIVE ensembles in the context of other studies. Black line indicates median values; grey shading likely range; unfilled bars 5-95% range. Studies included are: Nijsse et al. (2020); Tokarska et al. (2020); Jiménez-de-la Cuesta and Mauritsen (2019); Sherwood et al. (2020). CMIP6 indicates climate sensitivities derived from the energy balance model fits calculated above (including ocean heat uptake efficacy), CMIP6* indicates climate sensitivities derived using the Gregory method (Gregory et al., 2004) over the first 150 years of the abrupt-4xCO2 experiment.





### 515 4.3.3 Correlations between climate sensitivities and ERF

There are significant correlations between key variables in the CONSTRAINED ensemble, consistent with previous studies (Smith et al., 2018; Millar et al., 2015; Sanderson, 2020; Forest et al., 2002; Marvel et al., 2016). These are shown in the contour plots in figure 7.





**Figure 7.** Corner plot depiction of FULL and CONSTRAINED ensembles. Diagonal plots show marginal probability density functions of each key variable; FULL shown in grey, CONSTRAINED in black, and FULL constrained by the current level of warming only in red. Subdiagonal plots show contour plots of gaussian kernel density estimates of joint probability density. Contours show indicate regions containing 95, 67, 33 and 5 % of the ensemble members. Purple crosses and lines indicate the positions of individual CMIP6 models. The temperature and ERF variables are based on following an SSP2-45 trajectory from 2014 onwards.





### 4.3.4 Sensitivity to prior response parameter distributions

Previous work has shown that posterior marginal distributions of ECS and TCR depend strongly on the assumed prior distributions (Bodman and Jones, 2016). Here we test the sensitivity of our CONSTRAINED results to the response parameters sampled in FULL by replacing the TCR and RWF sample distributions stated in 4.2.3 with: $TCR \sim \mathcal{U}(0.5, 3.5)$ and $RWF \sim \mathcal{U}(0.2, 0.85)$. The actual prior distributions of TCR and RWF differ slightly from those stated here due to the rejection of unphysical response parameter sets, which tends to occur more often for lower values of TCR and higher values of RWF:

the quantiles of the input TCR and ECS distributions are 2.24 [0.87 , 3.37] and 4.11 [1.44 , 10.55] respectively. The posterior distributions of TCR and ECS after applying the constraint (ALTERNATIVE) described in 4.1 are 1.69 [1.15 , 2.54] and 3.14 [1.74 , 7.42]. The resulting marginal posterior distributions are wider than in the CONSTRAINED ensemble; though not considerably so for the TCR estimate. The upper end of the ALTERNATIVE ECS distribution is most affected by the change in prior, suggesting that the current level and rate of warming does not provide an exceptionally tight constraint on the upper

bound of the ECS. The ALTERNATIVE TCR distribution is not significantly different from CONSTRAINED, differing only by 0.1 K over the range of the distribution, demonstrating the close relationship between the TCR and historical warming (Sanderson, 2020) that enforces a tight constraint even with a significantly less informed prior. An analogous figure to figure 7 but for the ALTERNATIVE ensemble is in the Supplement (figure S3).

### 4.4 Constrained idealised experiments

Here we carry out standard CMIP6 experiments used in diagnosing the key properties of the climate - the abrupt-4xCO2 and 1pctCO2 experiments - with the FULL and CONSTRAINED parameter ensembles. This represents a test of whether our parameter sampling methods are sufficient to ensure that the range of carbon cycle and climate system responses are sampled from (as informed by the CMIP6 ensemble). We see in fig 8a, b that the FULL 90% credible interval closely matches the CMIP6 model ensemble range. The most significant discrepancy between the FULL ensemble and the CMIP6 model ensemble

is apparent in figures 8d, e, f; the FULL ensemble carbon cycle does not completely span the range of 11 C4MIP models, particularly at the high end of airborne fraction. This occurs due to the use of a lower central $r_0$ value, consistent with historical observations, in the FULL parameter sample, rather than taking the central value as the mean of the C4MIP model tunings. The FULL ensemble does span the range of carbon-cycle radiation feedback behaviour found in the C4MIP models (Arora et al., 2019, fig 8g). The CONSTRAINED ensemble, as expected from the climate sensitivity results above, is significantly

less spread than the CMIP6 model ensemble. It precludes both models with high and low climate sensitivities. Although our constraint does not significantly affect the carbon-cycle parameters, it does preclude some FULL ensemble members with a high airborne fraction, more apparent towards the end of the experiments. The CONSTRAINED ensemble implies a likely range (fig 8c) for the (CO2-only) TCRE (Matthews et al., 2009; Allen et al., 2009; Zickfeld et al., 2016; MacDougall, 2016) of 1.17 - 1.81, with a central estimate of 1.45 and 5-95% range of 1.00 - 2.13 K TtC$^{-1}$, based on the temperature response at a

cumulative CO2 emission of 1000 GtC. If instead we estimate the TCRE at a cumulative emission of 3000 GtC, this estimate is reduced by around 12% due to the slight non-linearity in the temperature–cumulative emission relationship. These estimates



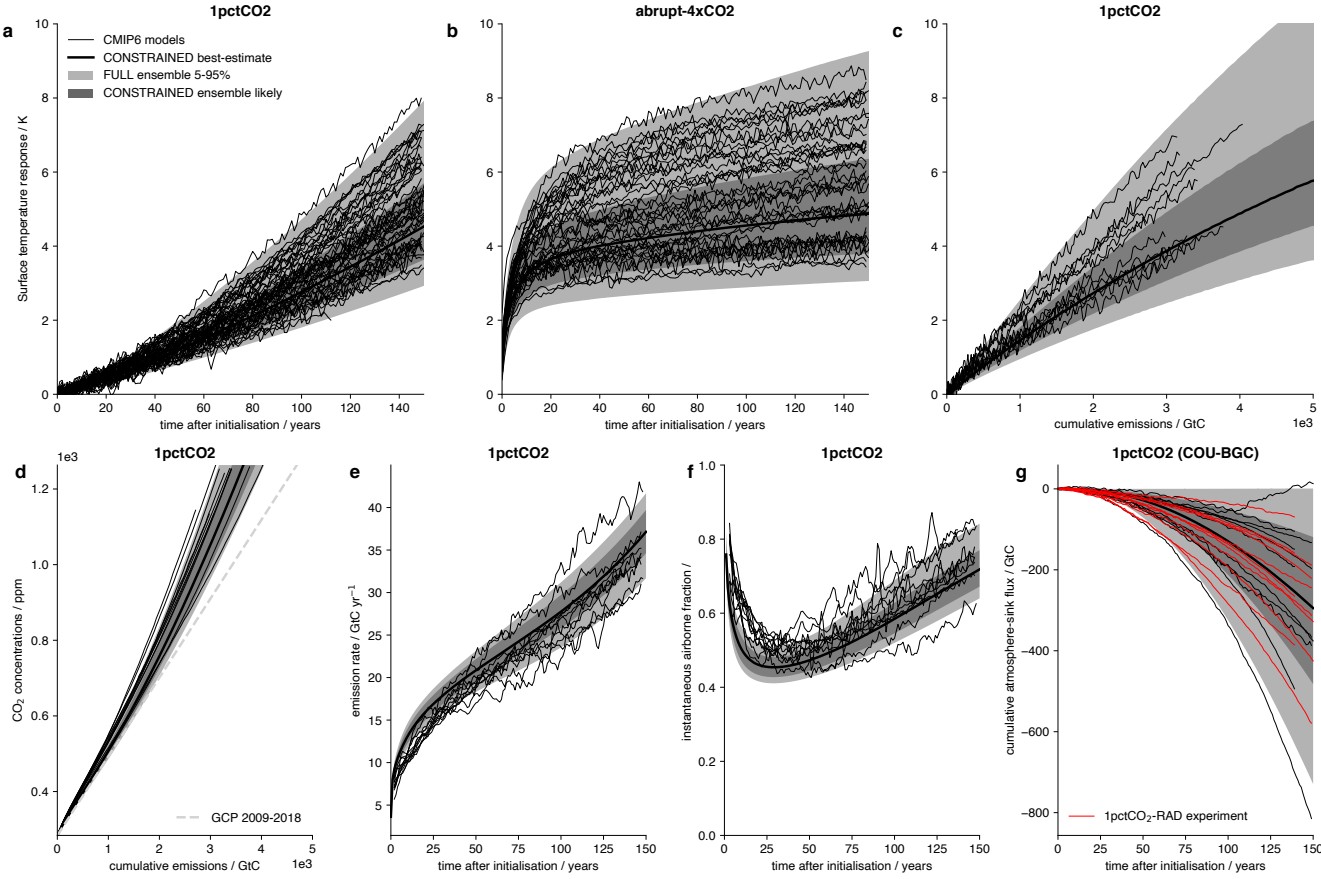

**Figure 8.** Idealised CMIP6 experiments with FULL and CONSTRAINED FaIRv2.0 ensembles. Thin black lines show drift-corrected CMIP6 model data. Light-grey shading indicates FULL ensemble 5-95% range. Dark-grey shading indicates CONSTRAINED ensemble 17-83% range. Thick black line shows central CONSTRAINED series. Dashed grey line in d shows airborne fraction for the most recent decade estimated from the Global Carbon Project data (Friedlingstein et al., 2019). Thin red lines in g show data directly from the radiatively-coupled C4MIP experiment, while thin black lines show an estimate of the radiation feedback on carbon sink strength as the difference between the fully- and biogeochemically-coupled C4MIP experiments.

are consistent with recent estimates based on the observational record (Millar and Friedlingstein, 2018; Gillett et al., 2013); though our best-estimate is slightly higher and the range less spread. This tighter range may be due to the noise reduction from using an idealised experiment and model with no representation of internal variability.





### 4.5 Constrained scenario projections

We use our CONSTRAINED parameter ensemble to project end-of-century warming and ERF in FaIRv2.0. Taking $CO_2$ emissions until 2014 from the Global Carbon Project (Friedlingstein et al., 2019), all other emissions from the RCMIP protocol, and land-use change and natural ERF timeseries from the SSP ERF timeseries database (Smith, 2020), we run both the FULL and CONSTRAINED ensembles for each SSP (Riahi et al., 2017). The FULL ensemble should not be used for projection and is included only to demonstrate the impact of the constraint.

| | FULL | | | | | CONSTRAINED | | | | |
| --- | --- | --- | --- | --- | --- | --- | --- | --- | --- | --- |
| 2100 warming relative to 1861-1880 / K | 5 | 17 | 50 | 83 | 95 | 5 | 17 | 50 | 83 | 95 |
| ssp119 | 0.749 | 1.02 | 1.49 | 2.10 | 2.67 | 0.848 | 1.00 | 1.27 | 1.63 | 1.96 |
| ssp126 | 0.984 | 1.33 | 1.92 | 2.65 | 3.32 | 1.100 | 1.29 | 1.63 | 2.05 | 2.44 |
| ssp245 | 1.550 | 2.08 | 2.97 | 4.03 | 4.93 | 1.800 | 2.08 | 2.56 | 3.12 | 3.60 |
| ssp370 | 2.150 | 2.91 | 4.18 | 5.65 | 6.84 | 2.720 | 3.08 | 3.67 | 4.31 | 4.81 |
| ssp370-lowNTCF-aerchemmip | 2.460 | 3.25 | 4.53 | 6.01 | 7.19 | 2.790 | 3.20 | 3.89 | 4.67 | 5.29 |
| ssp370-lowNTCF-gidden | 2.060 | 2.74 | 3.86 | 5.17 | 6.23 | 2.370 | 2.72 | 3.32 | 4.01 | 4.55 |
| ssp434 | 1.150 | 1.55 | 2.24 | 3.10 | 3.87 | 1.360 | 1.57 | 1.93 | 2.38 | 2.78 |
| ssp460 | 1.790 | 2.39 | 3.39 | 4.57 | 5.56 | 2.100 | 2.41 | 2.93 | 3.53 | 4.02 |
| ssp534-over | 1.140 | 1.54 | 2.24 | 3.14 | 3.96 | 1.280 | 1.50 | 1.90 | 2.41 | 2.88 |
| ssp585 | 2.880 | 3.80 | 5.29 | 6.99 | 8.33 | 3.250 | 3.73 | 4.54 | 5.46 | 6.18 |
| 2100 anthropogenic ERF / W m$^{-2}$ | | | | | | | | | | |
| ssp119 | 1.700 | 1.89 | 2.16 | 2.45 | 2.67 | 1.760 | 1.93 | 2.18 | 2.45 | 2.64 |
| ssp126 | 2.260 | 2.50 | 2.85 | 3.22 | 3.50 | 2.330 | 2.55 | 2.87 | 3.20 | 3.45 |
| ssp245 | 3.710 | 4.14 | 4.73 | 5.33 | 5.79 | 3.960 | 4.30 | 4.80 | 5.33 | 5.71 |
| ssp370 | 5.240 | 6.10 | 7.18 | 8.20 | 8.92 | 6.020 | 6.55 | 7.34 | 8.16 | 8.75 |
| ssp370-lowNTCF-aerchemmip | 6.190 | 6.79 | 7.65 | 8.53 | 9.19 | 6.440 | 6.97 | 7.74 | 8.54 | 9.11 |
| ssp370-lowNTCF-gidden | 5.120 | 5.70 | 6.54 | 7.40 | 8.05 | 5.380 | 5.88 | 6.63 | 7.39 | 7.95 |
| ssp434 | 2.640 | 2.96 | 3.41 | 3.86 | 4.21 | 2.850 | 3.10 | 3.46 | 3.85 | 4.13 |
| ssp460 | 4.320 | 4.80 | 5.48 | 6.18 | 6.69 | 4.630 | 5.01 | 5.58 | 6.17 | 6.60 |
| ssp534-over | 2.500 | 2.78 | 3.19 | 3.64 | 3.99 | 2.590 | 2.85 | 3.22 | 3.62 | 3.92 |
| ssp585 | 7.260 | 7.95 | 8.95 | 9.97 | 10.70 | 7.530 | 8.15 | 9.06 | 10.00 | 10.70 |



**Figure 9.** ERF timeseries by category for a range of SSP pathways using the CONSTRAINED parameter ensemble. Solid lines indicate central estimate and shading shows the 5-95% range. Dashed lines show default projection from a development version of the MAGICC SCM.







**Figure 10.** Surface temperature response projections for a range of SSPs with the CONSTRAINED parameter ensemble. Solid lines indicate central projection. Shading indicates a 5-95% range. Dashed line indicates default projection from a development version of the MAGICC SCM. Dots show the mean of 5 observational datasets. Bars on the right-hand side of the figure show end-of-century (2081-2100) warming. Filled bars show CONSTRAINED best-estimate, and likely and 5-95% ranges. Unfilled bars show CMIP6 median, and likely and minimum–maximum range. The number of CMIP6 models used in each scenario is given in table S4.





## 5  The response of simple climate models

The IPCC Special Report on 1.5°C warming (IPCC, 2018) included results from two SCMs, FaIRv1.3 (Smith et al., 2018) and MAGICC6 (Meinshausen et al., 2011a). One point of discussion following the report was the difference in results between these two models, with FaIRv1.3 tending to project a lower temperature response than MAGICC6 (Huppmann et al., 2018). This has resulted in a widely-held belief that FaIRv1.3 is intrinsically "cooler" than MAGICC6 in general, a belief that some of these authors have unintentionally contributed to previously (Leach et al., 2018). This belief is unfounded: the response of an SCM is a function of the parameters used. Although some parameters may be chosen to be consistent with geophysical observation or theory, in general SCM parameters are tuned such that they emulate, or reproduce, either the output of more complex models, or observations of the Earth. Relating this to the models used in SR15, the FaIRv1.3 ensemble was tuned such that the model response lay within observed changes in global mean surface temperature since pre-industrial (Smith et al., 2018; Cowtan and Way, 2014); the MAGICC6 ensemble was constrained to observations up until 2009 (Meinshausen et al., 2009). The two different tuning targets naturally leads to differences in the response of FaIRv1.3 and MAGICC6. Here we emphasize that the differences between the models' output is not systematic – it is the parameters used, and how these are selected (which is often a subjective decision on the part of the modellers), that determines the model response.

## 6  Uses of FaIRv2.0

We envisage that FaIRv2.0 will primarily be used for similar assessments as are carried out with the current SCMs, such as providing probabilistic projections of atmospheric concentrations, radiative forcings and temperature anomalies for wide ranges of scenarios, such as in the SR15 scenario explorer (Huppmann et al., 2018). FaIRv2.0 could also easily be coupled to integrated assessment models (IAMs) to explore impacts of climate policy options. One advantage that FaIRv2.0 has it that it was built with performance in mind, hence is easily vectorised. It can be vectorised in a programming language designed for array operations (such as Fortran, MATLAB, or the NumPy Python module) and hence FaIRv2.0 is extremely quick to run. For example, using its present Python implementation, FaIRv2.0 can compute the 1 million member FULL ensemble (emission driven for 52 gases, 78 forcing components, over the period 1750-2100) in under 30 minutes [2]. This speed provides significant advantages when computing large probabilistic ensembles, or when optimizing parameters. An important consideration for users computing probabilistic ensembles will be the memory required by FaIRv2.0 output, as this is more likely to be the limiting factor on a modern computer, rather than the model runtime. A related point is that FaIRv2.0 can be run in programs for analysis of tabular data, including (but not limited to) Excel. This opens up climate system exploration to a large group of potential new users who are familiar with spreadsheets, but not formal scientific programming languages.

---

[2]on a laptop with 31GB RAM and an Intel(R) Core(TM) i7-8750H@2.2GHz, 12 cores





We suggest that the speed, simplicity and transparency of FaIRv2.0 lends it to use in undergraduate and high-school education in addition to scientific research. It can be used to explain (and demonstrate) important features of both the carbon

(or other GHG) cycle and Earth's thermal response to radiative forcing, and is simple enough to use that students could themselves carry out experiments (such as a $CO_2$ doubling) easily with no prior experience and only basic computing skills.

FaIRv2.0 can also be used to rapidly investigate differences between ESMs, tuning FaIRv2.0 to emulate different full models and comparing differences between the tuned parameter sets to identify which aspects of the models differ most, as

was done with MAGICC in Meinshausen et al. (2011a, b). The ability to tune FaIRv2.0, as demonstrated here and in other work (Tsutsui, 2017; Joos et al., 2013; Millar et al., 2017), to more complex models also allows estimation of complex model response to a particular scenario or experiment without having to expend computer power to run the model itself; which could allow climate system uncertainties to be introduced more fully into integrated assessment studies by emulating the full CMIP6 ensemble within IAMs (providing some of the capability demonstrated by Meinshausen et al. (2011a) with a simpler model).


## 7   Conclusions

In this paper we have presented a significant update to the FaIRv1.3 SCM (Smith et al., 2018), focussed on reducing the structural complexity of the model as much as possible. The updated model, FaIRv2.0, uses the five equations of the AR5

impulse response model (Myhre et al., 2013) plus just one additional equation to allow the model to represent non-linearities in the carbon-cycle. We demonstrate that this reduction in complexity does not come at the cost of the model's ability to reproduce globally-averaged observations or output of more complex models from CMIP6 (Eyring et al., 2016). After demonstrating the ability of the model in emulating more complex models, we show how the model can be used for climate projection by constraining a large parameter ensemble.


There are many potential uses for FaIRv2.0 as a result of its simplicity and transparency. In addition to being available for the same probabilistic scenario assessment as is carried out by SCMs in reports such as SR15 (IPCC, 2018), it could be very easily implemented into IAMs; and likely improves computational efficiency due to its vectorisation and resulting extremely rapid runtime. We encourage policy-makers to use FaIRv2.0 in order to directly assess whether warming implications

are aligned with the intended outcomes of mitigation policies; since GHG accounting metrics used at present such as GWP do not provide accurate results for targets such as Net-Zero $CO_2$ due to the short life of some GHGs (Allen et al., 2018). To aid this use of FaIRv2.0, we will provide an Excel file containing the model with its default parameter set, ensuring FaIRv2.0 as available for all interested parties, even those unfamiliar with computer programming languages. The Excel version of the model could also be used to assist teaching of climate change and climate processes; and could even allow students

access to an easy-to-understand model that they could use themselves to explore future scenarios and the relative impacts

of future emissions of different greenhouse gases; or demonstrate the importance of climate sensitivity in an interactive manner.

We aim to be able to provide complete gas-cycle parameter sets tuned to each of the CMIP6 ESMs in the near future, such that FaIRv2.0 is able to be used to emulate the full CMIP6 ensemble and to improve understanding of how these models
differ in a single consistent framework. We hope that such parameter sets may also provide users in other related fields, such as climate policy, who do not have experience with climate models, access to a robust emulation of complex climate models with little learning curve.

*Code and data availability.* The code used to produce the figures is publicly available at https://github.com/njleach/GIR. However, we stress that the code here is *not* a model release. This update to the FaIR model will be made available at https://github.com/OMS-NetZero/FAIR
when fully integrated and tested. All data used in this study is publicly available at the relevant cited sources.

*Author contributions.* NJL, SJ and MRA conceived the study. NJL and SJ wrote the model code. NJL tuned the model to CMIP6 data and carried out the constrained ensemble. BW and TW assisted with tuning model parameters. CJS provided CMIP6 aerosol forcing data from RFMIP and advised on the forcing component parameterisations. JT advised on the thermal response component. JL and MC advised on model uses and tested the model. NJL produced the figures. NJL, CJS, ZN, SJ, JL and MRA wrote the manuscript.

*Competing interests.* The authors declare that they have no competing interests.

*Acknowledgements.* We acknowledge the World Climate Research Programme, which, through its Working Group on Coupled Modelling, coordinated and promoted both CMIP5 and CMIP6.



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
