# Peer review of "FaIRv2.0.0: a generalised impulse-response model for climate uncertainty and future scenario exploration"

_Geoscientific Model Development, 2020_

## Short Comment (SC1) · 21 Dec 2020

Dear authors,

as executive Editor of GMD I am writing this comment, as I do not understand your statement in the code availablity section:

"The code used to produce the figures is publicly available at https://github.com/njleach/GIR. However, we stress that the code here is not a model release. This update to the FaIR model will be made available at https://github.com/OMS-NetZero/FAIRwhen fully integrated and tested. All data

used in this study is publicly available at the relevant cited sources."

You anyhow describe the current version of the code, so according to GMD rules you have to make this version, which you are describing in this article, permanently available. This is independent from the point, that you can guide the reader to better use the final, fully integrated, more tested version of the code. But the idea of GMD is to enabling the direct relation between written publication and the code.

Therefore, in order to publish this article, provide the version described in this article in a permanent archive.

Best regards, Astrid Kerkweg

---

## Referee Comment (RC1) · Glen Peters (Referee) · 4 Jan 2021

The paper is well written and the analysis thorough. The model is suitably described and justified. The model will serve a useful scientific purpose, and should be easy for others to use (and potentially implement). With a nice interface, and the possibility to easily vary parameters, this model could become quite well used by the scenario community.

I generally have minor comments, and I see no major impediment to publication. A few of my comments go into quite some details, though I suspect they do not require much work to address (except maybe one or two depending on the response).

[Figure]

I list my comments in the order they appear in the manuscript:

1. Line 45: This sort of gives the impression that there are only two main SCMs, FaIR and MAGICC, and perhaps a few others. You sort of get there in the end, mentioning a "wide range of SCMs" from Nicholls et al, but this paragraph seems to really downplay the existence of other SCMs. I would suggest rewriting more in the narrative "There are dozens of SCMs that have been in widespread usage for decades... Scenarios generated by IAMs and assessed by the IPCC have generally used one SCM (MAGICC), with FaIR additionally used in IPCC SR15. Something about RCMIP, etc." Of course, choose your own words and framing, but this paragraph gives a very different impression than reality would have it!

2. Line 51: This is more a passing comment. FaIR might be five equations, etc, but that does not mean it is simple and transparent. And computing resources are not such a limitation for many SCM problems these days, so a more process based SCM can have advantages even if it has more equations and takes longer to solve. I don't see the main application or even motivation of FaIR to be some IPCC SCM. FaIR was used in SR15, was available for AR6, but has not exactly taken IPCC by storm. Maybe AR7... At the end of the day, all SCMs require parameters and calibration, even FaIR has hundreds of parameters (Table S2), and there are dozens of ESMs to tune to, alternative ways to implement various processes, various ways to tune, etc, so there will always be debate about whatever SCM is used and how it is parameterised. Long story short, I would see the motivation of FaIR to do good and exciting science. If FaIR is used as a harmonising SCM across IPCC WGs, I would see that as a co-benefit (but not a motivation).

3. Figure 1: It would be useful somewhere to give equations for Gu and Ga. Also mention somewhere (caption) that terms without (t) are constant for each gas. Also, probably worth mentioning (caption), that that n=1 for most species. This figure in a way over complicates the simplicity!

4. Figure 1: It would be useful to put in numbers in each box to emphasise the sequencing ("model steps take place from left to right"), but noting, this sequencing will relate also to implementation. There is not much written about implementation, but the sequencing implies this is solved using forward differences? This is ultimately a system of differential equations, and they don't need to be solved by forward difference, so the sequencing is an artifact of the implementation?

5. Line 114: Would it be fair to say this feedback approach is a fudge? To use a non-technical term. . . This is ok, as FaIR does a good job at replicating ESMs, so don't take this as a critique. What are the physical interpretations of these r coefficients? Everything gets wrapped into alpha, and it becomes hard to disentangle. I note also that the 100 is fixed, and not given as a parameter. Does this mean that 100 is essentially arbitrary, and I could use 5, 50, and 500 years and get the same results (with different r coefficients)? Does the choice of 100 years change key parameters or just the r coefficients? For example, if I used 500 years would the TCR be different or would feedbacks be adequately represented? I am ok with the approach, just need a few more words on the implications. Would it be better, or even desirable, to have the 100 as a free floating parameter that can be determined to get the best fit?

6. Line 125: I would explicitly mention that g0 and g1 are constants for each gas, as it is not really that apparent on the first read. Also, specify the equations for Gu and Ga.

7. Line 125 and line 150 (section). I understand that this is all generalised, but as Table S2 shows, most of the parameters are zero (or 1). CO2 and CH4 are exceptions. It may make sense for Equation (1) to say, here is the generalised form necessary for CO2, and here is the specific form for n=1 which applies to each other gas. What is alpha for everything other than CO2 and CH4? Is it just 1 (was too hard to do in my head)? A bit more information here for CO2, CH4, and all others is also important for implementation. If it is known that the alpha term is not used for most species, implementation could be different (CO2 is treated as an exception). In any case, a bit more emphasis should be put on the formulation being that everything is somewhat a

structural subcase of CO2.

8. Figure 2: Is Halon1202 correct? Seems a way off observations?

9. Line 210+: So, to be clear, I just put in anthropogenic emissions, and FaIR takes care of the rest? I don't need to specify natural emissions? (it is sort of one of these things I want to test!)

10. Section 2.2: As for the concentration equations, the general form is only needed for CO2, CH4, and N2O? Worth mentioning the simplifications otherwise it all looks so complex. . .

11. Line 319: Table S5 has the parameters C, kappa, etc. (it took me a while to find these). The C, kappa, etc, are the physical ones that are more important than d and q? Do these values make physical sense? I see in the table some values differ quite some from others. Do these differences indicate something with the calibration or other ESM specific issues? It also looks like C3 can take on many values? d3 seems to not add much value to the times, it could be 70 years to 1000s of years. Since the experiments used for calibration have short run times, is there really enough data to fit three exponentials? Time scales of 1000s of years in a fit over a few hundred is meaningless. Perhaps two exponentials are more than sufficient? Dropping the third exponential may not change the quality of the fit. Was there a reason to take 3 terms?

12. Line 319: I don't think the paper gives default values for the climate system? Table S2 has all the default values for the atmospheric response and forcing, but what are the default climate parameters?

13. Section 2 overall: There is nothing mentioned about implementation. It would be useful to write something about how you implemented this. Was it a simple forward difference (as implied by Figure 1)? Did you calculate all equations for all components, or simplify for non-CO2 and non-CH4 species? You note how fast the code solves, but from experience, fast code usually requires smart implementation. Another key

aspect for solution times is the time step. If you used an explicit method, the time step would need to satisfy some constraints. I think of these as all quite important points to discuss, if one hopes to implement the code... And you note one key advantage of the model is that it is so simple and easy for others to implement...

14. Table 2: Each column seems to have different significant digits? Is there a rationale?

15. Figure 3: I guess the figure is from 1850 (axis label not that great)

16. Figure 4: What happens with GFDL? FaIR seems to have variability?

17. Table 5: How would these parameters change if 100 years was not used for the feedback? What about 50 years, 200 years, etc?

18. Section 4: You often write "are given below" when it should say "are given in Table X" or something.

19. Figure 7: The 2010-2019 warming seems like the key constraint? On a 10 year time scale, variability could still be at play, so this constraint is locking in some variability? Would it be better to take a climatology of say 30 years? What happens if this constraint is removed altogether? You did some analysis for the alternative ECS priors, but I think it makes sense to remove some of these constraints to see how they effect the solution.

20. Figure 8d: It is a bit unclear what the reference to GCP is here? Is this the period over which the airborne fraction is calculated? Might need a word or two extra...

21. Section 4.5: Does the Table miss a caption?

22. Section 4.5, Table: For ssp119/26, does it make sense to show two additional rows for peak warming?

23. Figure 10: By eye, it looks like MAGICC and FaIR differ by around 0.2°C (for the high and low pathways)? This is not trivial for 1.5°C! And the difference to CMIP6 is rather significant. This comes all the way back to my 2nd comment... How do I determine if MAGICC or FaIR is a better representation? Maybe MAGICC performs better across all scenarios, maybe not, but how would I assess this? If MAGICC performs better, would I use FaIR if it only has five equations and solves in a microsecond? And is the performance related to structural issues or parameterisations? Maybe FaIR would perform better overall with some tweaks to the calibration? There is also the question of whether the task of FaIR is to represent the CMIP6 ensemble, or to represent a constrained version, or both (of course). I know this figure is illustrating the model, but it opens many questions in the context of how the paper was framed. In any case, these are all points for the discussion! And these points are why it is so nice to have such a "simple and transparent" SCM, as you can do so many quick analyses to answer these sorts of rather fundamental questions!

24. One issue not mentioned is the paper is additivity and nonlinearities. It is not uncommon to want to know what the effect of a given sector or gas is? How much is CO2, or SO2, or transport, or China, etc of the total? In that case, it would be nice to know how non-linear the model is. If I ran all the components separately (Table S2) and added together, how close to the total is it? What about if I added sectors or countries, would the sectors and countries add to the global total? If not, how big or important is the difference? Since you hope this model will be widely used, and even has an EXCEL version, the additivity and nonlinearity issues really need to be discussed and quantified.

25. Just to mention again, it would be good to have a section where the implementation is discussed. The paper mentions this can run in EXCEL and is easy for others to implement, but just running through my head, it doesn't necessarily seem completely trivial! It would be great to explain how it was implemented and how the solution time was optimised given time steps, etc. You mention there are also memory issues too, so mentioning these sorts of issues is important if people want to implement the model. It would also be useful for implementation purposes to provide some standardised input and output data, so people can test their implementation and problem solve any

implementation errors.

26. Table S2: I guess that is a r_tau, not tau?

27. Table S2: Great if this data, and similar data for the temperature model, are easily to download as csv or something as these are key parameters for people to implement the model.

---

## Referee Comment (RC2) · Anonymous Referee #2 · 12 Feb 2021

This manuscript presents a major update of the FaIR model, it clearly explains the main equations of the model, and presents parameterizations for a set of GHGs and forcings. As I understand, this manuscript has gone already through a major round of reviews and revisions, and this version is already in a very advanced stage. I do not have major comments, and I think it can be accepted after a few minor revisions.
**1 Minor comments**

- Sec 2.1, second line (line numbers do not add up. Very likely a misuse of LaTeX line numbers with equations). I would rather call it a '4-timescale IRF' than a '4-pool IRF'. You can think about an IRF as a coordinate transformation, that takes a four-pool carbon cycle model and maps it to a four coordinate system along four eigen directions with respective eigenvalues. Also, I assume you are talking here only about IRFCO2 from Joos et al. (2013), and not the other IRFs in their Table 5. Please clarify.
- Eq. 1. The time-dependency in the adjustment factor is missing. You should write  $\alpha(t)$ .
- Ln 145. Isn't more appropriate to say 'carbon dioxide' than 'carbon cycle'? For CO2 n=4, and for methane n=1, so CO2 includes the full complexity of the approach, but not methane.
- Ln 177. Please check the units of the pre-industrial CH4 concentration, ppb instead of ppm?
- Section 2.4. The state-space representation of the temperature response is a very interesting an elegant way to express these equations. However, I do not think it is correct to include the forcing term *F* as part of the vector of states. It doesn't have the same units as the temperature variables, and it is a non-autonomous term. I suggest expressing this equation as

$$\dot{\mathbf{X}} = \mathbf{F}(t) + A\,\mathbf{X} \tag{1}$$

 $\mathbf{F}(t) = \begin{pmatrix} F(t)/C_1 \\ 0 \\ 0 \end{pmatrix}$ (2) C2 GMDD
with

and

$$A = \begin{pmatrix} -(\lambda + k_2)/C_1 & k_2/C_1 & 0\\ k_2/C_2 & -(k_2 + \epsilon k_3)/C_2 & \epsilon k_3/C_2\\ 0 & k_3/C_3 & -k_3/C_3 \end{pmatrix}$$
(3)

GMDD
In this representation, you obtain a matrix A that is invertible, which would guarantee that you can perform an eigen decomposition on the entire matrix, and not just on a portion of it, as expressed in lines 323-325. Also, it better expresses the fact that in this model, temperatures respond to a time-dependent forcing according to a set of fixed timescales and heat capacities of a three-box model.

---

## Author Comment (AC1) · 11 Mar 2021

Thank you very much for your comment. Upon submitting our revision, we will publish all the code used to create the figures and carry out the analysis within this manuscript in a permanent repository. This includes code for the version of the model (FaIRv2.0.0-alpha) used within this manuscript.

Best wishes, Nicholas Leach & co-authors

---

## Author Comment (AC2) · 11 Mar 2021

Firstly, thank you for a thorough and extremely helpful review. Your comments and suggestions, particularly those relating to the carbon cycle explanation and the paper conclusions, have significantly improved the manuscript. Below are our point-by-point responses to you comments.

1. This is a very good point, and definitely not the impression we would want to give. We have re-phrased this paragraph along the lines suggested.

2. This is entirely fair. Similar to your later suggestion (#23), we feel that this discussion

of compromise when it comes to SCMs would be a good addition to our conclusion section. As suggested, we believe FaIR is very close to as simple as you can get without losing some emulation ability (or similarly to real Earth System response). However, this comes at the cost of a reduced connection to physical processes. Hence for some applications in which processes are crucial, FaIR won't be adequate. This being said, for much of the work done on scenario assessment, FaIR would be adequate - and possibly preferable as it is only five equations hence others could reasonably re-implement it if they wish to test outputs without relying on our implementation. In summary, your points are well taken, and hopefully we have reflected them in our revised conclusions.

3. This is a great suggestion and we've added in equations for Gu and Ga into the first equation set. We have also amended the caption in line with both your other suggestions.

4. Yes, this is correct. We will (as suggested) include some notes regarding implementation in the supplement. The sequencing does certainly suggest forward differences, as we have used - and probably also hints at what languages this particular author is most familiar with!

5. Short answer - yes, this feedback approach is something of a fudge (as an analytic approximation of the Millar et al. parameterisation which explicitly used the iIRF100). On the question of the impact of the 100-year timescale, under the FaIRv2.0 formulation, changing this to eg. 5/50/500 years would simply change the values of g0,g1 & the r coefficients (for say an optimal fit to a particular C4MIP model), and would not fundamentally change the behaviour at all - the optimal fit should be independent of this timescale choice. One motivation behind keeping it at 100 years would be for consistency with the previous iterations of the FaIR model (since the approximation is very good, r coefficients are easily comparable between previous iterations and those provided in this paper).

6. This is a very helpful suggestion - we have changed the text accordingly.

7. This is a fair comment. The main reasoning behind only including the general case was largely because this is how we've implemented it (returning to the point you make about including something on implementation). The way we'd recommend implementing is (at least for an array-focussed language like python or matlab) to use the fully general form and vectorise the calculation over all the gases, as this is likely optimal even if there are lots of redundant elements of 1s and 0s; it also makes the code somewhat more compact and (arguably) easier to understand since you don't have to write out all the different cases. Alpha is indeed (effectively) 1 for all the other gases. I say effectively because for all the other gases, we've set alpha equal to 100 & tau_1 equal to the actual lifetime / 100 (resulting in alpha * tau being equal to the actual lifetime). This is because for exceptionally long-lived gases, the g_0 & g_1 calculations become unstable (not representable within float precision), so we use this "trick" to get around this numerical issue. We'll make sure to try and emphasize how everything is really just a simpler (subcase) of CO2 better in the revision.

8. As you suspect, this is a mistake. Halon1202 emissions are zero in the RCMIP emission dataset for the SSPs, hence the flat FaIRv2.0.0 and MAGICC7.1.0 lines. We therefore exclude it from this figure, but still include it in the default parameter set.

9. Yes - in FaIRv2.0.0 we haven't parameterised in any sort of natural emissions (explicitly, though a quantity of natural emissions is implied by the pre-industrial concentration parameter). The real motivation behind this is transparency (& consistency with the CO2 gas cycle). If you want to include changes to the natural emissions of a particular GHG, then these would have to be included as a residual (over pre-industrial natural emissions) in addition to any anthropogenic emissions you put into the model.

10. This is correct - except that we actually don't even use the general form for N2O. We have re-worded parts of this section to try and emphasize that the equations underlying the majority of the gas cycles could be re-written in a simpler format than the general form given (ie. without an alpha parameter).

11. Yes - the parameters in S5 correspond to the more physically motivated parameters set out in (eg.) Geoffroy et al., 2013. The extreme-high values of C3 (& d3) do appear to represent fits in which the 3rd timescale adds little to the fit; and for these models it is highly likely that two exponentials would provide an identically good fit. The motivation behind using a three-exponential form (as opposed to the two-exponential form in previous FaIR versions) was drawn from Cummins et al., 2020 and Tsutsui, 2020. These two papers both suggested that using three exponentials provided significantly improved emulation relative to two. Summarising, the reasoning behind Cummins et al., 2020 and Tsutsui, 2020 was as follows. Improved response to large forcing impulse with three timescales (since the shortest timescale can be shorter), as the emulation can respond more rapidly to a sharp change in forcing, such as in the abrupt-4xCO2 experiment or a volcanic eruption as observed in GCMs. Cummins et al. obtained optimal emulation with three timescales rather than two for all 16 CMIP5 models they investigated.

12. This is correct, and we have amended this in the revision, taking default climate response parameters from the central estimate of the CONSTRAINED ensemble (as this would represent our best-estimate of the actual climate system response).

13. This is a very useful suggestion, and we now include a section in the Supplement about our implementation (python-focused, though should apply more generally).

14. This is an error on our part. All the numeric tables now specify quantities to three significant figures.

15. Yes- this is amended in the revision (for the idealised experiments we now measure time relative to initialising the model in year zero).

16. GFDL had variability in the CO2 concentration data (we are not certain why), which projected onto the FaIRv2.0.0 diagnosed emissions. Before, we smoothed the temperature series to obtain relatively noise-free diagnosed emissions. However, in the revision, we do not pre-process the raw data at all, resulting in diagnosed emissions

with significant variability inherited from the variability of the temperature series input (and CO2 concentration in the case of GFDL). We have done this for transparency, but if you feel it confuses the message of the figure then we can revert to smoothing the temperature & CO2 inputs?

17. See reply to comment 5 above.

18. This is very helpful - we've changed the text accordingly throughout.

19. Yes - the warming level is a stricter constraint than the rate. In the revision we've combined the two so they can no longer be imposed separately, making disentangling the contributions more tricky. The time period used in the constraint should not make too much difference as the global warming index is noise-free (though uncertainty due to internal variability is still incorporated) due to the underlying forced signal being noise-free. We now include a section on the sensitivity of the constraint to the observational dataset used in the GWI computation (in addition to the sensitivity to the priors), which hopefully helps to clear up some of the questions you have here.

20. We will amend the caption to clarify this. It is meant to represent the 2010-19 airborne fraction implied by the "anthropogenic carbon flows" figure (#9) in the global carbon budget paper.

21. Yes - unsure what's gone wrong here, but possibly a LaTeX compiling problem. . . will ensure the revision doesn't have this issue.

22. This is a great suggestion! We've added in extra rows for the scenarios that peak before 2100 (SSP1-19, 1-26 & 5-34).

23. These are some tricky questions - but nonetheless important, of course. We aren't very keen to make any suggestions of which model is "better"; and don't want to overlap too much with the much more comprehensive SCM comparisons in the RCMIP paper (Nicholls 2020a, b). We attempted to address some of these issues in the "response of simple climate models" section, though we agree that it would be good to have more

of a discussion around these issues in the "Conclusions" section. We have added an additional paragraph in the "Conclusions" which addresses some of the pros / cons of FaIR (& other SCMs) and how they might be used. On the point of whether FaIR is intended to represent the CMIP6 ensemble or a constrained version, we certainly agree that the answer is both. The CMIP6 representations could be useful for understanding how ESMs differ within the context of a very simple and transparent parameterisation. On the other hand, the constrained version is certainly more relevant in terms of scenario assessment and policy (especially in light of the quite high response of several CMIP6 models). We had attempted to outline this view in the "Uses of FaIR" section, but have re-worded this section to make these thoughts clearer.

24. We have now included a section in the supplement concerning the nonlinearities in FaIR. Since nonlinearities are introduced primarily through the CO2 and CH4 gas cycles, we focus on a series of CO2 / CH4 pulse experiments (against an SSP2-45 reference scenario). Although this doesn't completely characterise the nonlinearities (since by definition they are pathway-dependent), we think that it provides at the least some qualitative information about how the nonlinearities arise in FaIR; and also about some limiting cases for the default FaIR parameterisation, such as the response under a very large methane pulse ($\sim$1000GtCO2-eq), where the methane lifetime increases considerably and therefore significantly changes the behaviour of the model to CH4 emissions.

25. We've now added a brief implementation section outlining our own python implementation in the supplement.

26. Very good spot! Yes - completely correct and we've amended this in the revision.

27. This is a great idea. We're creating a permanent archive for the analysis (figures, model etc.) used here, and this will include a .csv of these key parameters, which we can point people to in the table caption.

Best wishes, Nicholas Leach & co-authors

References:

Nicholls, Z., Lewis, J., Makin, M., Nattala, U., Zhang, G. Z., Mutch, S. J., ... Meinshausen, M. (2021). Regionally aggregated, stitched and de‐drifted CMIP‐climate data, processed with netCDF‐SCM v2.0.0. Geoscience Data Journal, 00, gdj3.113. https://doi.org/10.1002/gdj3.113

Thornhill, G. D., Collins, W. J., Kramer, R. J., Olivié, D., Skeie, R. B., O'Connor, F. M., ... Zhang, J. (2021). Effective radiative forcing from emissions of reactive gases and aerosols – a multi-model comparison. Atmospheric Chemistry and Physics, 21(2), 853–874. https://doi.org/10.5194/acp-21-853-2021

Skeie, R. B., Myhre, G., Hodnebrog, Ø., Cameron-Smith, P. J., Deushi, M., Hegglin, M. I., ... Wu, T. (2020). Historical total ozone radiative forcing derived from CMIP6 simulations. Npj Climate and Atmospheric Science, 3(1), 1–10. https://doi.org/10.1038/s41612-020-00131-0

Haustein, K., Allen, M. R., Forster, P. M., Otto, F. E. L., Mitchell, D. M., Matthews, H. D., & Frame, D. J. (2017). A real-time Global Warming Index. Scientific Reports, 7(1), 15417. https://doi.org/10.1038/s41598-017-14828-5

Geoffroy, O., Saint-Martin, D., Olivié, D. J. L. L., Voldoire, A., Bellon, G., Tytéca, S., ... Tytéca, S. (2013). Transient Climate Response in a Two-Layer Energy-Balance Model. Part I: Analytical Solution and Parameter Calibration Using CMIP5 AOGCM Experiments. Journal of Climate, 26(6), 1841–1857. https://doi.org/10.1175/JCLI-D-12-00195.1

Cummins, D. P., Stephenson, D. B., & Stott, P. A. (2020). A new energy-balance approach to linear filtering for estimating effective radiative forcing from temperature time series. Advances in Statistical Climatology, Meteorology and Oceanography, 6(2), 91–102. https://doi.org/10.5194/ascmo-6-91-2020

Tsutsui, J. (2020). Diagnosing Transient Response to CO 2 Forcing in Coupled Atmosphere‐Ocean Model Experiments Using a Climate Model Emulator. Geophysical Research Letters, 47(7). https://doi.org/10.1029/2019GL085844

Nicholls, Z., Meinshausen, M., Lewis, J., Gieseke, R., Dommenget, D., Dorheim, K., ... Xie, Z. (2020). Reduced complexity model intercomparison project phase 1: Protocol, results and initial observations. Geoscientific Model Development Discussions, 1–33. https://doi.org/10.5194/gmd-2019-375

Nicholls, Z. R. J., Meinshausen, M. A., Lewis, J., Rojas Corradi, M., Dorheim, K., Gasser, T., ... et al. (2020). Reduced Complexity Model Intercomparison Project Phase 2: Synthesising Earth system knowledge for probabilistic climate projections. Earth and Space Science Open Archive, 29. https://doi.org/10.1002/ESSOAR.10504793.1

Millar, R. J., Nicholls, Z. R., Friedlingstein, P., & Allen, M. R. (2017). A modified impulse-response representation of the global near-surface air temperature and atmospheric concentration response to carbon dioxide emissions. Atmospheric Chemistry and Physics, 17(11), 7213–7228. https://doi.org/10.5194/acp-17-7213-2017

---

## Author Comment (AC3) · 11 Mar 2021

Thank you very much for your comments and suggestions. We have taken them all into account in our revised text. Our point-by-point responses to your comments are given below.

1 (On the carbon cycle IRF). These are good suggestions. The "pool" nomenclature used follows Millar et al., 2017. In the revision, we have used your suggestion of calling this carbon cycle model a "4-timescale IRF" where possible. When referring to the model carbon stores (the $R_i$), we use the term "reservoirs", and are explicit that these are not physical carbon stores when they are introduced.

[Figure]

2 (alpha time dependency). Thank you very much for pointing out this error. We have changed the text to correct this.

3 (carbon dioxide vs carbon cycle). This is a good suggestion. We have changed the text accordingly.

4 (CH4 pre industrial units). You are correct, and the text has been changed accordingly.

5 (EBM matrix representation). This is a good suggestion, and we have re-written the equation set accordingly.

Best wishes, Nicholas Leach & co-authors

References:

Nicholls, Z., Lewis, J., Makin, M., Nattala, U., Zhang, G. Z., Mutch, S. J., . . . Meinshausen, M. (2021). Regionally aggregated, stitched and de‐drifted CMIP‐climate data, processed with netCDF‐SCM v2.0.0. Geoscience Data Journal, 00, gdj3.113. https://doi.org/10.1002/gdj3.113

Thornhill, G. D., Collins, W. J., Kramer, R. J., Olivié, D., Skeie, R. B., O'Connor, F. M., . . . Zhang, J. (2021). Effective radiative forcing from emissions of reactive gases and aerosols – a multi-model comparison. Atmospheric Chemistry and Physics, 21(2), 853–874. https://doi.org/10.5194/acp-21-853-2021

Skeie, R. B., Myhre, G., Hodnebrog, Ø., Cameron-Smith, P. J., Deushi, M., Hegglin, M. I., . . . Wu, T. (2020). Historical total ozone radiative forcing derived from CMIP6 simulations. Npj Climate and Atmospheric Science, 3(1), 1–10. https://doi.org/10.1038/s41612-020-00131-0

Haustein, K., Allen, M. R., Forster, P. M., Otto, F. E. L., Mitchell, D. M., Matthews, H. D., & Frame, D. J. (2017). A real-time Global Warming Index. Scientific Reports, 7(1), 15417. https://doi.org/10.1038/s41598-017-14828-5

[Figure]

Geoffroy, O., Saint-Martin, D., Olivié, D. J. L. L., Voldoire, A., Bellon, G., Tytéca, S., . . . Tytéca, S. (2013). Transient Climate Response in a Two-Layer Energy-Balance Model. Part I: Analytical Solution and Parameter Calibration Using CMIP5 AOGCM Experiments. Journal of Climate, 26(6), 1841–1857. https://doi.org/10.1175/JCLI-D-12-00195.1

Cummins, D. P., Stephenson, D. B., & Stott, P. A. (2020). A new energy-balance approach to linear filtering for estimating effective radiative forcing from temperature time series. Advances in Statistical Climatology, Meteorology and Oceanography, 6(2), 91–102. https://doi.org/10.5194/ascmo-6-91-2020

Tsutsui, J. (2020). Diagnosing Transient Response to CO 2 Forcing in Coupled Atmosphereâ ĂŘOcean Model Experiments Using a Climate Model Emulator. Geophysical Research Letters, 47(7). https://doi.org/10.1029/2019GL085844

Nicholls, Z., Meinshausen, M., Lewis, J., Gieseke, R., Dommenget, D., Dorheim, K., . . . Xie, Z. (2020). Reduced complexity model intercomparison project phase 1: Protocol, results and initial observations. Geoscientific Model Development Discussions, 1–33. https://doi.org/10.5194/gmd-2019-375

Nicholls, Z. R. J., Meinshausen, M. A., Lewis, J., Rojas Corradi, M., Dorheim, K., Gasser, T., . . . et al. (2020). Reduced Complexity Model Intercomparison Project Phase 2: Synthesising Earth system knowledge for probabilistic climate projections. Earth and Space Science Open Archive, 29. https://doi.org/10.1002/ESSOAR.10504793.1

Millar, R. J., Nicholls, Z. R., Friedlingstein, P., & Allen, M. R. (2017). A modified impulse-response representation of the global near-surface air temperature and atmospheric concentration response to carbon dioxide emissions. Atmospheric Chemistry and Physics, 17(11), 7213–7228. https://doi.org/10.5194/acp-17-7213-2017

---

## Author Comment (AC4) · 11 Mar 2021

We thank both Glen Peters and one anonymous reviewer for their comments and suggestions. We have attempted to address each comment in turn, with point-by-point responses in the separate response documents. In addition to our responses to the specific comments made by the reviewers, we have also updated the manuscript in a number of other places. These changes aim to ensure the text and model is fully up-to-date with the current state of the literature, and make several improvements to the transparency and robustness of the analysis within the manuscript. These additional changes are set out directly below.

We now use emission data solely from RCMIP throughout, for reasons of consistency and transparency.

We use CMIP6 data from https://cmip6.science.unimelb.edu.au/ where possible throughout, processed as set out in Nicholls et al., 2021. This includes the CMIP6 tunings.

Updates to the default parameterisation:

We have updated the ozone parameterisation in line with the recent studies from Thornhill et al., 2021, and Skeie et al., 2020. This parameterisation means we no longer distinguish between tropospheric & stratospheric ozone.

The default aerosol parameters are now set to be equal to the central values of the CONSTRAINED ensemble.

The default climate response is set to be equal to the central response of the CONSTRAINED ensemble.

We now include (where data is available) verification runs for the CMIP6 climate response emulation, using the abrupt-2xCO2 and abrupt-0p5xCO2 experiments as verification experiments for the tunings (which are computed using the abrupt-4xCO2 and 1pctCO2 experiments). However, these data are not available for all models we emulate, so we cannot verify the tuned parameters for each model.

We have updated the constraint methodology used to determine CONSTRAINED parameter sets from the FULL ensemble. Before, we used a simple pass/fail criterion based on the global warming index calculation (GWI, Haustein et al., 2017). While this was very straightforward, it did not make full use of the GWI, which provides an estimate of the distribution of current level and rate of warming; and additionally it sampled from regions of exceptionally low likelihood in level/rate space. In our updated constraint, we set the probability of selecting an individual ensemble member equal to the likelihood of the present-day level & rate of warming as determined by the GWI. This is

a significantly stricter criterion (only ∼8 % of the FULL ensemble is retained).

Within this new constraint methodology, in addition to the sensitivity to prior assumptions on climate response, we also test the sensitivity to the observational dataset used to compute the GWI (particularly relevant with the arrival of HadCRUT5). This demonstrates how future projections can be impacted by the choice of observational dataset used in such a constraining procedure. We believe that this additional test of sensitivity is particularly useful given the several recent papers that use observed warming to constrain future projections.

References:

Nicholls, Z., Lewis, J., Makin, M., Nattala, U., Zhang, G. Z., Mutch, S. J., ... Meinshausen, M. (2021). Regionally aggregated, stitched and de‐drifted CMIP‐climate data, processed with netCDF‐SCM v2.0.0. Geoscience Data Journal, 00, gdj3.113. https://doi.org/10.1002/gdj3.113

Thornhill, G. D., Collins, W. J., Kramer, R. J., Olivié, D., Skeie, R. B., O'Connor, F. M., ... Zhang, J. (2021). Effective radiative forcing from emissions of reactive gases and aerosols – a multi-model comparison. Atmospheric Chemistry and Physics, 21(2), 853–874. https://doi.org/10.5194/acp-21-853-2021

Skeie, R. B., Myhre, G., Hodnebrog, Ø., Cameron-Smith, P. J., Deushi, M., Hegglin, M. I., ... Wu, T. (2020). Historical total ozone radiative forcing derived from CMIP6 simulations. Npj Climate and Atmospheric Science, 3(1), 1–10. https://doi.org/10.1038/s41612-020-00131-0

Haustein, K., Allen, M. R., Forster, P. M., Otto, F. E. L., Mitchell, D. M., Matthews, H. D., & Frame, D. J. (2017). A real-time Global Warming Index. Scientific Reports, 7(1), 15417. https://doi.org/10.1038/s41598-017-14828-5

Geoffroy, O., Saint-Martin, D., Olivié, D. J. L. L., Voldoire, A., Bellon, G., Tytéca, S., ... Tytéca, S. (2013). Transient Climate Response in a Two-Layer Energy-Balance

Model. Part I: Analytical Solution and Parameter Calibration Using CMIP5 AOGCM Experiments. Journal of Climate, 26(6), 1841–1857. https://doi.org/10.1175/JCLI-D-12-00195.1

Cummins, D. P., Stephenson, D. B., & Stott, P. A. (2020). A new energy-balance approach to linear filtering for estimating effective radiative forcing from temperature time series. Advances in Statistical Climatology, Meteorology and Oceanography, 6(2), 91–102. https://doi.org/10.5194/ascmo-6-91-2020

Tsutsui, J. (2020). Diagnosing Transient Response to CO 2 Forcing in Coupled Atmosphere‐Ocean Model Experiments Using a Climate Model Emulator. Geophysical Research Letters, 47(7). https://doi.org/10.1029/2019GL085844

Nicholls, Z., Meinshausen, M., Lewis, J., Gieseke, R., Dommenget, D., Dorheim, K., ... Xie, Z. (2020). Reduced complexity model intercomparison project phase 1: Protocol, results and initial observations. Geoscientific Model Development Discussions, 1–33. https://doi.org/10.5194/gmd-2019-375

Nicholls, Z. R. J., Meinshausen, M. A., Lewis, J., Rojas Corradi, M., Dorheim, K., Gasser, T., ... et al. (2020). Reduced Complexity Model Intercomparison Project Phase 2: Synthesising Earth system knowledge for probabilistic climate projections. Earth and Space Science Open Archive, 29. https://doi.org/10.1002/ESSOAR.10504793.1

Millar, R. J., Nicholls, Z. R., Friedlingstein, P., & Allen, M. R. (2017). A modified impulse-response representation of the global near-surface air temperature and atmospheric concentration response to carbon dioxide emissions. Atmospheric Chemistry and Physics, 17(11), 7213–7228. https://doi.org/10.5194/acp-17-7213-2017

---

## Author Response (AR2)

Dear Dr. Sierra,

Thank you for your comments on our revised manuscript. We have now uploaded both the model code and notebooks used within the paper to Zenodo, and now provide the respective DOIs in the Code Availability Statement section.

Best wishes,
Nicholas Leach & co-authors

---

## Author Response (AR3)

We have updated the author affiliations to be correct at this time. We have
updated the paper acknowledgments to include all the appropriate funding sources
and thank the reviewers of the manuscript and revisions.

Best wishes,
Nicholas Leach & co-authors